# Characterisation of Atlantic Meridional Overturning hysteresis using Langevin dynamics

Jelle van den Berk[1], Sybren Drijfhout[1,2], and Wilco Hazeleger[2]

[1]Royal Netherlands Meteorological Institute, De Bilt, The Netherlands
[2]Faculty of Geosciences, Utrecht University, Utrecht, The Netherlands

**Correspondence:** J. van den Berk (jelle.van.den.berk@knmi.nl)

**Abstract.** Hysteresis diagrams of the Atlantic Meridional Overturning Circulation (AMOC) under freshwater forcing from climate models of intermediate complexity are fitted to a simple model based on the Langevin equation. A total of six parameters are sufficient to quantitatively describe the collapses seen in these simulations. Reversing the freshwater forcing results in asymmetric behaviour that is less well captured and appears to require a more complicated model. The Langevin model allows for comparison between models that display an AMOC collapse. Differences between the climate models studied here are mainly due to the strength of the stable AMOC and the strength of the response to a freshwater forcing.

## 1 Introduction

The Atlantic Meridional Overturning Circulation (AMOC) is an important circulation in the Atlantic ocean. It is also an important part of the climate system overall due to the heat it transports from the South Atlantic to the North Atlantic (Ganachaud and Wunsch, 2000; Vellinga and Wood, 2002). The AMOC therefore has a substantial influence on the (western) European climate and a weakening of the AMOC might cause changes in the European climate and weather. The AMOC has also been identified as one of Earth's 'tipping elements' where a rapid change on markedly faster times scales could take place in the (near) future (Lenton et al., 2008). The AMOC is partly buoyancy driven by the deep water formations in the North Atlantic subpolar gyre which produces the North Atlantic Deep Water (NADW) (e.g. Rahmstorf (2000)). The AMOC might be bi-stable in nature which means it admits an 'off' state, with little or no transport from north to south, as a counterpart to its current 'on' state (Broecker et al., 1985).

Palaeoclimate records of the last glacial period show a rapid switching of temperature, which might be associated with the presence/absence of a vigorous AMOC as exists today (Dansgaard et al., 1993). The possibility of a bistable AMOC being the cause of these rapid changes has been noted (Broecker et al., 1990). With the current climate warming rapidly, the stability of the AMOC is of particular interest (Collins et al., 2013) and climate modelling projections indicate the AMOC strength will decrease under an increase of $CO_2$. Recent measurements show the AMOC has decreased in strength (Smeed et al., 2018). An understanding of the possibly bistable nature of the AMOC is therefore relevant to understand the consequences of climate change. See Weijer et al. (2019) for a review on AMOC bistability.

The Langevin equation has been posited before as suitable to capture the essential dynamics of an AMOC collapse (Ditlevsen and Johnsen, 2010; Berglund and Gentz, 2002). It has also been used elsewhere as the basis for describing the dynamics of climate sub-systems (Kwasniok and Lohmann, 2009; Livina et al., 2010) and the AMOC in particular (Kleinen et al., 2003; Held and Kleinen, 2004). A fourth order potential function is used in Ditlevsen and Johnsen (2010); Berglund and Gentz (2002) because it is the minimum required for having three distinct solutions (double wells). This potential function has two parameters which are presumed to be functions of the freshwater forcing. Variation in the freshwater forcing is assumed to directly drive changes in AMOC strength by changing the potential function in the Langevin equation. Although the hysteresis loops of the AMOC include both a collapse and a resurgence point, we will only attempt to model the collapse from the stable 'on' branch to the stable 'off' branch.

Though the Langevin equation has played a role in the conceptual picture of bistability and tipping points in the climate, it has not been used to actually fit the parameters to a (simulated) AMOC collapse. Here, we attempt to construct a simple model based on the Langevin equation and fit its dynamics to salt-advection driven collapse trajectories of the AMOC seen in climate models (Rahmstorf et al., 2005). The result is a set of parameters that quantitatively describe the AMOC collapse process. This derived model defines a low-dimensional manifold that captures the essential AMOC collapse characteristics. To the extent that the low-dimensional model is successful in capturing the more complex model this method could also be used to predict the parameter range where in a model a collapse would occur. At present, however, it is intended to provide a characterisation of the collapse that will allow comparison between climate models.

Section 2 sketches the theoretical background of the Langevin equation and of the salt-advection mechanism. In Section 3 we fit the proposed Langevin model to the AMOC collapse trajectories seen in a set of climate models of intermediate complexity (EMICs) taken from Rahmstorf et al. (2005). We end with a discussion and conclusions in Section 4.

## 2   The Langevin model

An increase in surface air temperatures, or an increased surface freshwater flux by changes in precipitation minus evaporation, will decrease the buoyancy in the shallow layer of the deep water formation regions in the North Atlantic subpolar gyre. The deep water formation is reduced, and the southward meridional flow reduced. In principle, this mechanism can reduce the AMOC to zero gradually if fully buoyancy-driven. A salt-advection feedback mechanism that leads to a bimodal AMOC was proposed by Stommel (1961). In this mechanism, salinity anomalies in the North Atlantic are amplified by the overturning flow, which in turn controls the North Atlantic salinity. Positive anomalies are strengthened and negative anomalies weakened; this results in a positive feedback between the salinity anomalies and the overturning. Bistability, consisting of a strong and a weak AMOC state, and possible abrupt transitions result.

Fig. 1 shows a conceptual picture of the two stable AMOC (index) states. The AMOC is a scalar variable obtained by integrating the overturning transport and selecting its maximum value (typically located in the subtropical North Atlantic). In red, the upper branch is drawn up to the collapse point where a bifurcation occurs. The real AMOC in the current climate moves along this branch from the left, to the right, towards its (assumed) collapse point. The branch in blue is the counterpart

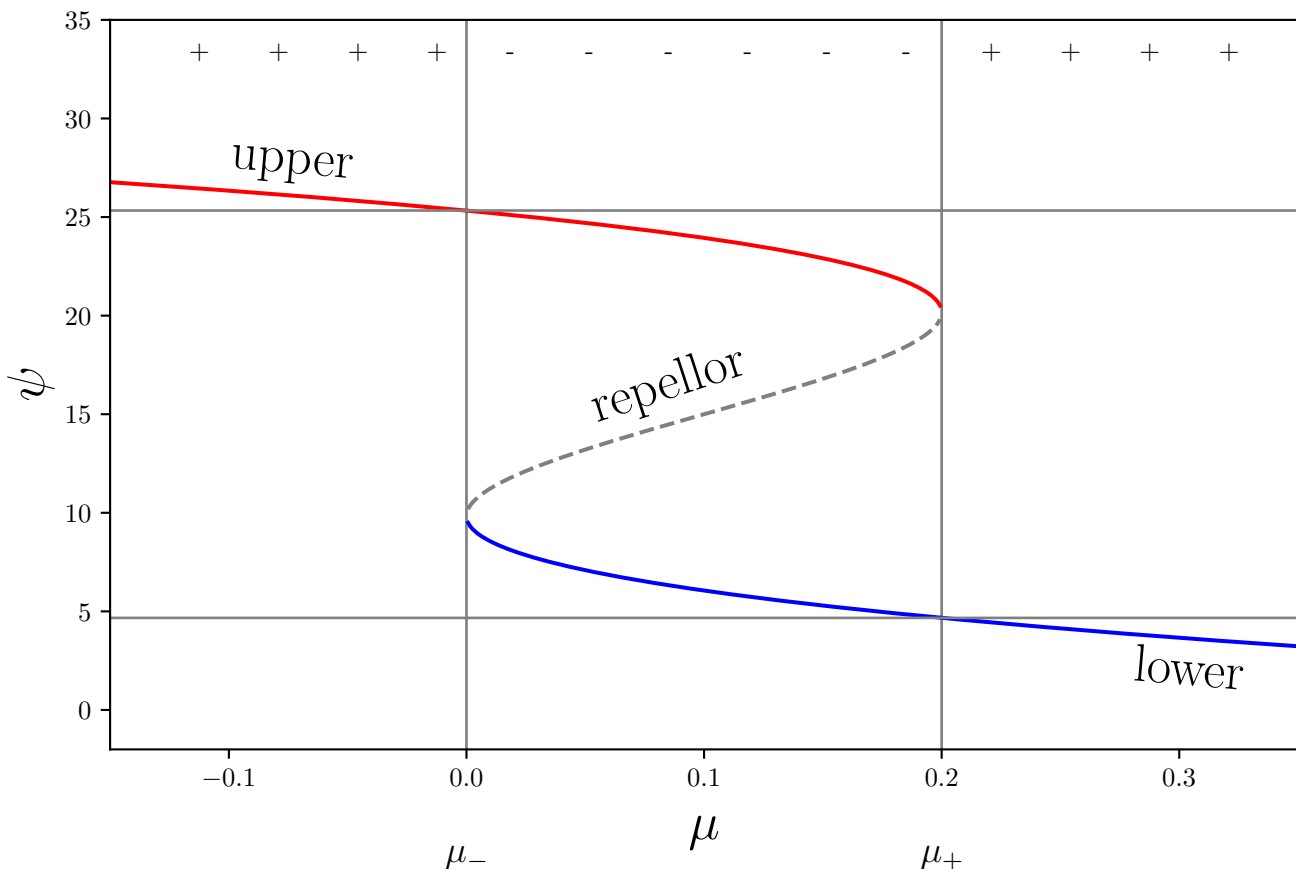

**Figure 1.** Example bifurcation diagram of the AMOC ($\Psi$) in response to a control variable $\mu$. The red branch is the on-state (upper), blue the off-state (lower). The upper branch deforms when closer to the bifurcation points which are connected though the repellor (dashed line). The two bifurcations points are indicated as $\mu_+$ (collapse point) and $\mu_-$ (resurgence point). Top $\pm$ symbols indicate unimodal ($+$) or bimodal ($-$) regime.

of the upper branch and represents the off state of the AMOC and ends in another bifurcation point to the left where the AMOC jumps back to full strength. The dashed line (repellor) separates the two basins of attraction associated with the two stable branches (attractors). At the bifurcation point one of the two basins of attraction vanishes and a qualitative change takes place in the potential function (the number of solutions for a given value of the freshwater forcing $\mu$ goes from 3 to 1).

Below we will derive a model based on the Langevin equation that captures the essential dynamics of a bimodal AMOC under a freshwater forcing $\mu$.

## 2.1 Multiple stable AMOC states

The conceptual picture of the AMOC being a zero-dimensional variable that is driven by stochastic forces trapped in a potential is similar to that of a particle's motion described by Langevin dynamics (Lemons et al., 1908). The Langevin equation (Gardiner, 2004; Ditlevsen and Johnsen, 2010),

$$\dot{x} = -\partial_x U_\mu(x) + \sigma\eta \tag{1}$$

describes the position of a noise-driven particle ($x$) trapped in a potential function $U$. The stochastic term is a white noise process ($\eta$) scaled with an intensity parameter $\sigma$. At first we will ignore the stochastic nature of the AMOC collapse process and focus on the deterministic behaviour.

The double well potential seen in Fig. 1 has been extensively studied and applied, also in a quantitative way. But to our knowledge it has not been quantitatively applied to AMOC hysteresis using the Langevin equation in complex numerical climate models before.

AMOC bistability has, however, been studied quantitatively in e.g. Boulton et al. (2014) using transient runs. In Poston and Stewart (1978) an extensive treatment is given why, in addition to a scaling and shifting, only two parameters are sufficient to describe the bistability. More precisely, the third order term and the fourth order coefficient can be eliminated. The two remaining coefficients in the polynomial describe the critical behaviour, not just locally near the critical points, but the entire trajectory under a suitable transformation. A direct consequence is that only partial information, in the form of a piece of the trajectory, should suffice to describe the entire trajectory (the full hysteresis loop).

The potential function takes the form (Gardiner, 2004; Ditlevsen and Johnsen, 2010)

$$-U(x) = -\frac{1}{4}x^4 + \frac{\beta}{2}x^2 + \alpha x. \tag{2}$$

The two parameters $\alpha, \beta$ are functions of the freshwater forcing $\mu$. The AMOC state variable $\Psi$ requires an affine transformation (Cobb, 1980),

$$\alpha = \alpha(\mu)$$
$$\beta = \beta(\mu)$$
$$x = (\Psi - \lambda)/\nu.$$

To fit the model trajectories we need to find expressions for $\alpha$ and $\beta$, and suitable values for the transformation parameters $\lambda$ and $\nu$. In the literature $\alpha$ is referred to as the normal factor, and $\beta$ the splitting factor (Poston and Stewart, 1978). In the bifurcation diagram the value of $\nu$ is approximately the distance in $\Psi$ between the bifurcation point on the top branch to the bifurcation point on the lower branch. Similarly, the value of $\lambda$ is approximately the $\Psi$ value between the bifurcation points at $\mu_\pm$. The transformation uses $\lambda$ to shift the trajectory and $\nu$ to scale it. Below we describe the potential visually and state additional constraints that follow from the demand that the freshwater forcing is the only variable that determines the dynamical behaviour.

## 2.2 Potential description

In Fig. 2 an overview of the qualitatively different forms of potential are shown ($-U(x)$, right panels) together with their derivative functions ($-\partial_x U$, left panels). Dots indicate the location of critical points and are related to the number of wells in the potential. The top panels show the typical bimodal form ($I$) with two stable states and one unstable one in the middle. Below these are the three possible unimodal states ($E$). These occur for forcing values to the left of $\mu_-$ and to the right of $\mu_+$. The panels $B_1$ and $B_2$ are the submanifolds that separates the unimodal regime from the bimodal regime. These two meet in the cusp point $P$, as shown in the bottom panels. See Poston and Stewart (1978) for further details.

In Fig. 3 the stability diagram is shown where the areas indicated are those with qualitatively different behaviour seen in Fig. 2. See also Poston and Stewart (1978) for similar diagrams. The cusp point $P$ is the singular point where no proper solution can exist because only the trivial solution (all parameters are valued 0) is allowed here (both bifurcation points $\mu_\pm$ and AMOC strength are at zero). The two parameters are $\alpha$ and $\beta$ and are the two coefficients in the potential function. Their values change because of their dependency on the forcing value ($\mu$).

Our aim is to arrive at a description that matches a series of $\mu$ values across the stability diagram. The two parameters $\alpha, \beta$ are independent but can be parameterised by other variables that map them to observations. If parameterised by a single variable, the values of ($\alpha, \beta$) across the stability surface are a one-dimensional subset, as suggested by the AMOC index. On one side of the cusp point, along the splitting axis ($\beta$), only a unimodal regime exists, while on the other side two regimes exist with the modes at relative distances apart.

## 2.3 Constraints

With a varying $\alpha$ there exist an interval between two critical points ($\alpha_\pm$) in between which the distribution is bimodal and unimodal outside that interval. Because the AMOC trajectory is 1-dimensional and $\mu$ is also 1-dimensional, there must be a relation between $\alpha$ and $\beta$ that reduces dimensionality from two to one dimensions. When passing through the critical point $\alpha_+$, the number of potential wells goes from two to one. Similarly, moving through $\alpha_-$ changes the number of wells from one to two (for given $\mu_\pm$). The two critical points of $\partial_x U$, $\mu_\pm$, can be found analytically for $\mu_\pm$ real and being degenerate solutions. It can be shown (Birkhoff and Mac Lane, 1970, p. 106) that the discriminant $D = 27\alpha^2 - 4\beta^3 = 0$ (i.e. real solutions) needs to be solved for $\alpha$ to obtain the two critical solutions that relate $\alpha$ and $\beta$. It is at these solutions that the number of critical points changes at forcing values $\mu_\pm$. (When $D < 0$ there are three distinct real solutions which corresponds to the bimodal regime,

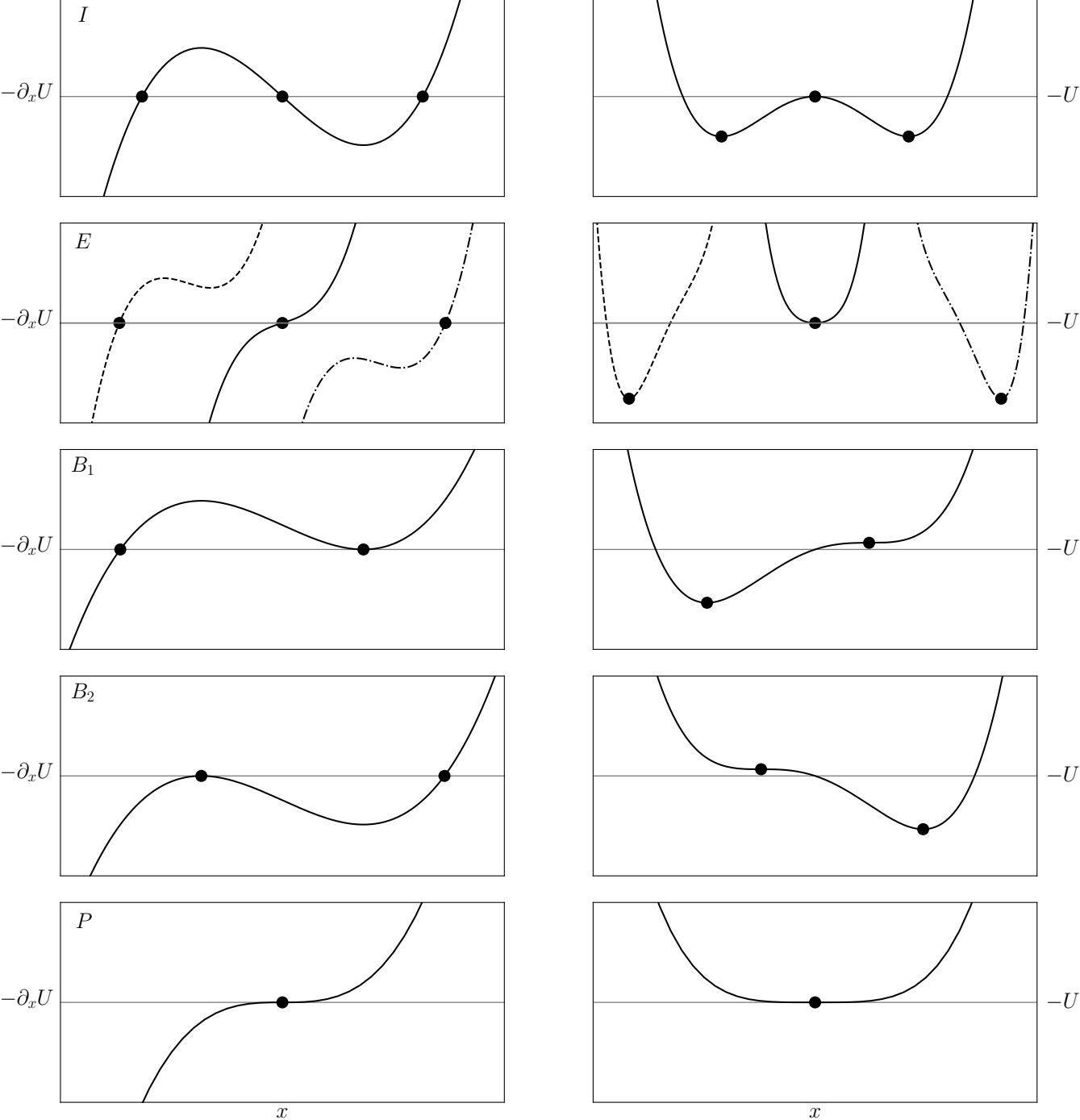

**Figure 2.** Sample potentials (right) and their derivatives (left) for (top to bottom) the three possible varieties of bimodal state ($I$), three types of unimodal state ($E$), the two pathological cases where $D = 0$ ($B_1$ and $B_2$), and the cusp catastrophe point ($P$). Dots indicate the critical points. (Scaling is not uniform between panels. Note the choice of negative sign of the potential $U$.)

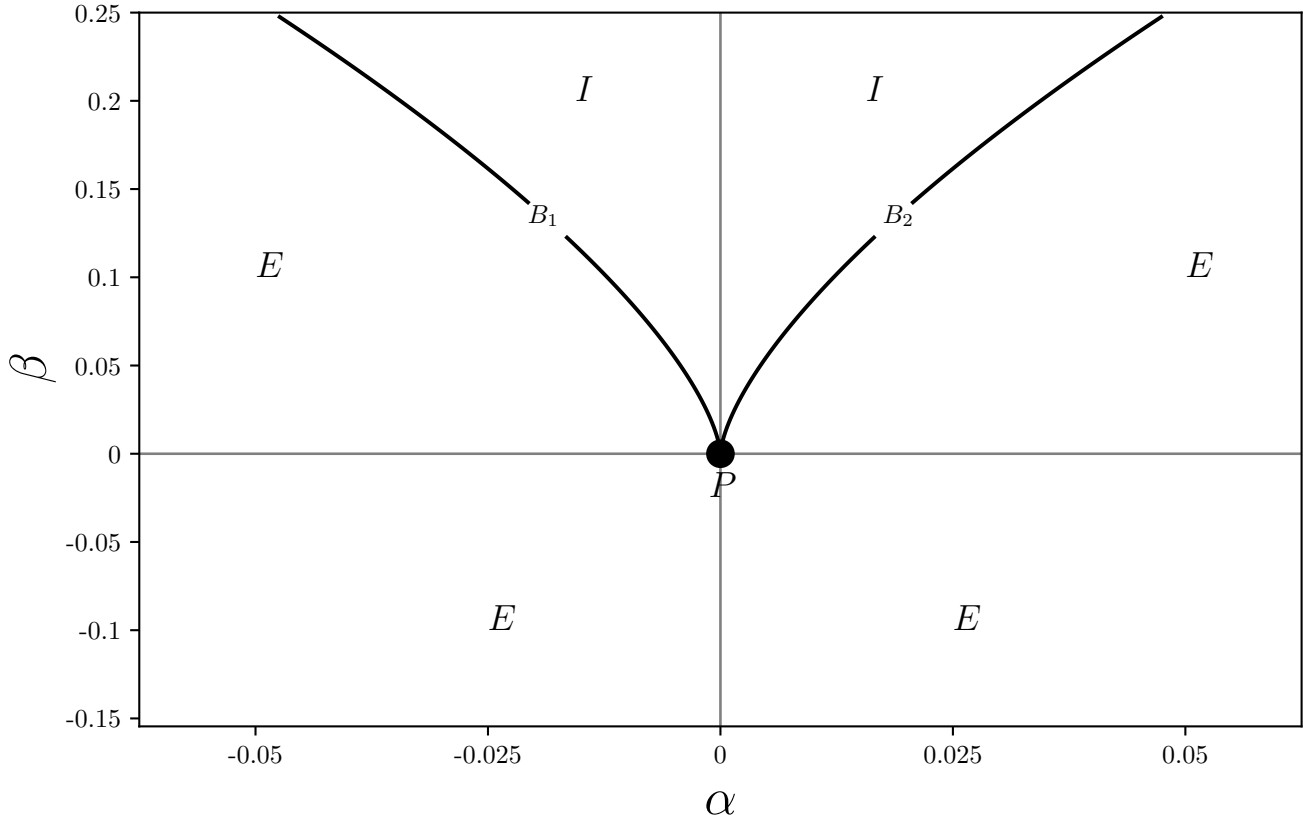

**Figure 3.** Discriminant determining the stability and number of critical points. The splitting factor $\beta$ and normal factor $\alpha$ describe the stability diagram. The bimodal regime ($I$) is separated from the unimodal regime ($E$) by two lines ($B_{1,2}$) which meet in the point $P$.

when $D > 0$ there is only one distinct real solution, which corresponds to the unimodal regime.) When any two of the roots are the same, the number of extrema goes from 3 to 2 (or 1 if all are the same) and the solutions become degenerate (this occurs at $B_{1,2}$ in Fig. 3).

Solving for $\alpha$ gives two solutions that are the critical values as functions of $\beta$,

$$\alpha_\pm = \pm \frac{2\sqrt{3}}{9} (\beta)^{3/2} \quad \text{or} \quad \alpha_\pm = \mp \frac{2\sqrt{3}}{9} (\beta)^{3/2},$$

with $\beta \geq 0$ for real solutions. The points $\alpha_\pm$ correspond to where the lines $B_{1,2}$ in Fig. 3 are passed when moving across the stability surface.

For $\alpha_+ < 0 - U(1) < 0$. This corresponds with the AMOC undergoing a collapse at $\mu_+$ from an on state to an off state, and the correct choice of sign is

$$\alpha_\pm = \mp \frac{2\sqrt{3}}{9} (\beta_\pm)^{3/2}, \tag{3}$$

with $\alpha_{\pm}$ and $\beta_{\pm}$ the values corresponding to $\mu_{\pm}$. Changing $\mu$ in the bifurcation diagram corresponds to moving from curve $B_2$ to curve $B_1$ and Eq. 3 relates the two stability parameters $\alpha$ and $\beta$ at the two critical forcing values $\mu_{\pm}$.

### 2.3.1 Linear functions $\alpha, \beta$

The value of $\beta$ does not need to be fixed (to $\alpha_{\pm}$ and in general there is a corresponding $\beta_{\pm}$ at the respective critical points. We assume linear functions for $\alpha$ and $\beta$,

$$\alpha(\mu) = \alpha_0 + \mu \, \delta\alpha$$

$$\beta(\mu) = \beta_0 + \mu \, \delta\beta,$$

reducing the dependency to these four parameters. Linear functions are the simplest non-trivial dependencies, while adding non-linear parameters introduces further unknowns, making this the most parsimonious parametrisation that captures the first order behaviour. Also, intuitively we can understand the pair $(\delta\alpha, \delta\beta)$ as the angle under which the system moves to the bifurcation point ($B_{1,2}$) in Fig. 3), which locally only requires the values of $\alpha$ and $\beta$ up to first order. From this parametrisation we can determine the offset $\alpha_0$ and rate $\delta\alpha$ in terms of $\beta_0$ and $\delta\beta$,

$$\alpha_+ = \alpha_0 + \mu_+ \delta\alpha = -\frac{2\sqrt{3}}{9} \left(\beta_+\right)^{3/2} \quad \text{and}$$

$$\alpha_- = \alpha_0 + \mu_- \delta\alpha = +\frac{2\sqrt{3}}{9} \left(\beta_-\right)^{3/2}$$

gives

$$\delta\alpha = -\frac{2\sqrt{3}}{9} \frac{\left(\beta_+\right)^{3/2} + \left(\beta_-\right)^{3/2}}{\mu_+ - \mu_-} \tag{4}$$

$$\alpha_0 = \alpha(\mu = 0) = \frac{\sqrt{3}}{9} \left[ -\left(\beta_+\right)^{3/2} + \left(\beta_-\right)^{3/2} \right] - \frac{1}{2} \delta\alpha \left(\mu_+ + \mu_-\right). \tag{5}$$

This constrains the values of $\alpha$, leaving only $\beta$ as a free variable, which is then parameterised by $\beta_0$ and $\delta\beta$. Note that only solutions with $\beta_{\pm} > 0$ are valid. Also, values for $\beta_0$ and $\delta\beta_0$ that result in crossing $B_2$ in another point besides $\beta_-$ are unsuitable. (The curves $B_{1,2}$ are each intersected by a straight line in at most two points, and we require intersection at a single point only.)

### 2.4 Stochastic interpretation

With the deterministic framework in place, the stochastic nature can be reintroduced. The potential function can be replaced by a distribution which is the stationary distribution in the asymptotic limit (i.e. the long term behaviour of repeated sampling of the hysteresis loop). The potential (a fourth-order polynomial) gives the probability distribution (Cobb, 1978)

$$P(x, \alpha, \beta) = C e^{-2/\sigma^2 \, U(x)} = C e^{2/\sigma^2 \left(-1/4 x^4 + \beta/2 x^2 + \alpha x\right)}. \tag{6}$$

The factor $C = C(\alpha, \beta)$ does not have a (known) analytical expression for the general case, but can be computed numerically (and can therefore used as a factor in the likelihood function in the next section). This can be done accurately with an adaptive

quadrature method (Piessens et al., 2012), though it suffers from numerical limitations. The value of $\sigma$ is a measure of intrinsic variation in the AMOC. Note that $\sigma$ is a measure of additive noise (because we assume that $\sigma$ is not dependent on $\mu$) and

160 other choices, such as multiplicative noise, can be made (Das and Kantz, 2020). See Gardiner (2004) for a derivation of this distribution using the Fokker-Planck equation, from which also the Langevin equation can be derived. Also, note that $\sigma \to \sigma/\nu$ because of the scaling with $\nu$ we introduced in Section 2.1.

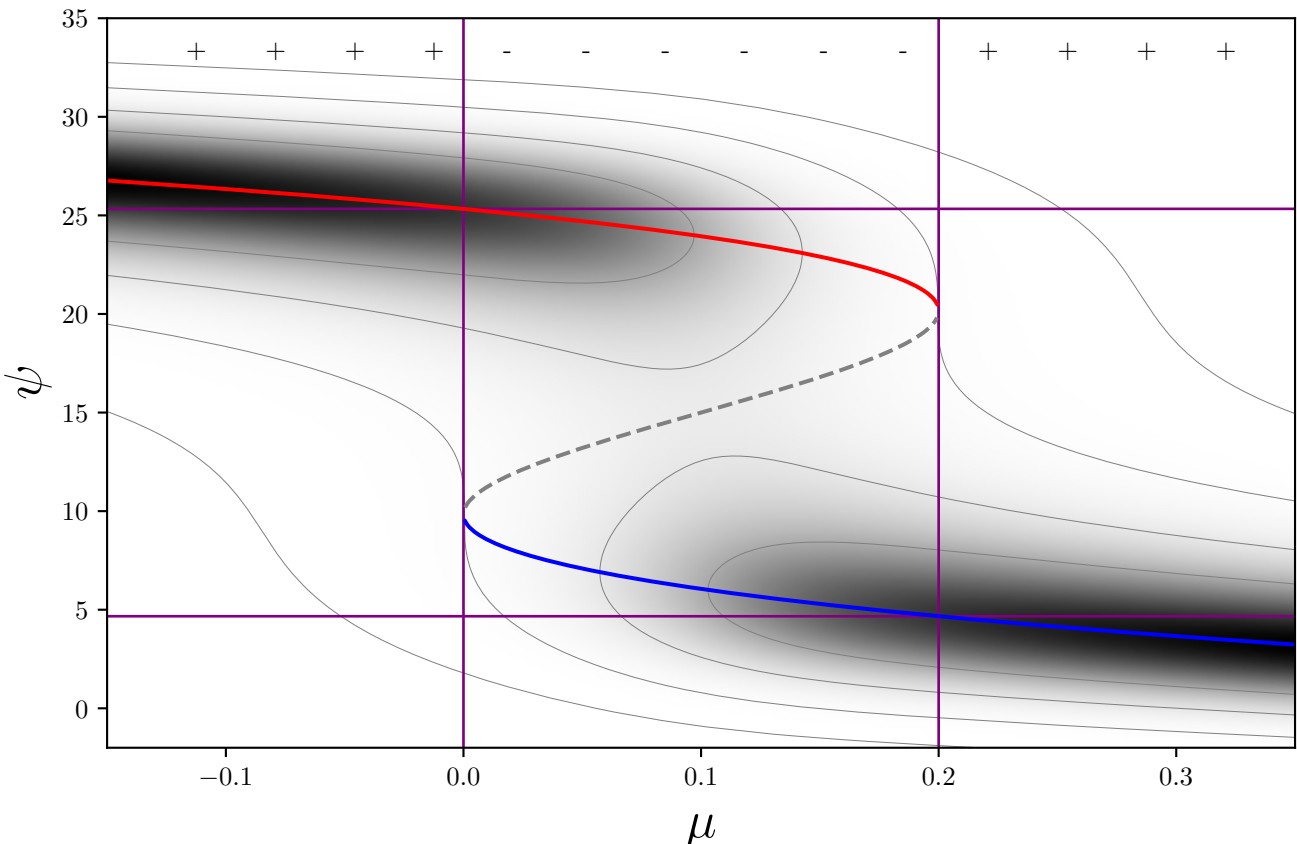

**Figure 4.** Example trajectory with corresponding distribution. Parameterised by $\lambda = 15$, $\nu = 20$, $\sigma = 0.12\nu$, $\mu_+ = 0.2$, $\mu_- = 0$, $\beta_0 = 0.2$, $\delta\beta = 0$; $\alpha$ under constraints in Eqs 4 and 5. The distribution of one of the attractor branches (red: on state, blue: off state) deforms when closer to the bifurcation points which are connected though the repellor that forms the trench of the distribution (dashed line). Top $\pm$ symbols indicate unimodal ($+$) or bimodal ($-$) regime based on the discriminant value ($D$). The value of $\sigma$ is relatively large and is chosen for clarity. The purple lines indicate the (fixed) positions of the bifurcation points.

An example bifurcation diagram with corresponding distribution is shown in Fig. 4. The purple lines indicate the (fixed) positions of the bifurcation points. The dashed grey line marks the positions of the unstable solution (repellor) in between the

165 two attractor branches which separates the two basins of attraction. Note that the bifurcation points are extremal in the sense that no bimodality can exists beyond them. With the trajectories being noisy and driven along the attractor, there is (always)

some probability of a 'noise-induced' transition. The state shifts from one basin of attraction to the other, crossing the repellor, and the AMOC rapidly moves from one attractor to the other. For this reason, the bimodality region might be larger than is apparent from a particular sample AMOC trajectory. A larger noise level (as seen in AMOC observations Smeed et al. (2018)) would increase the likelihood of a collapse before the AMOC reaches the bifurcation point.

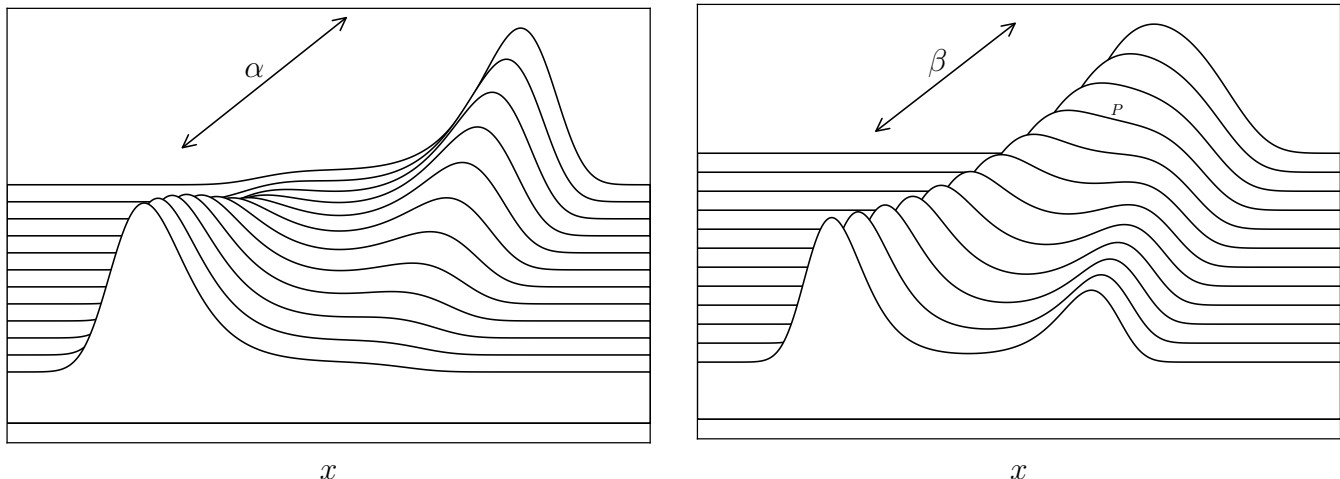

**Figure 5.** Left: Distributions from the exponential family (Eq. 6) where the parameter $\beta$ is kept at a fixed value and $\alpha$ is varied. The distribution transforms from unimodal (back), to bimodal (middle), to a different unimodal distribution (front). The bimodal states have a larger and a smaller mode, depending on the position within the bimodal regime. The relative strength between modes depends on $\sigma$. Right: Distributions from the exponential family (Eq. 6) where the parameter $\alpha$ is kept at a fixed value and $\beta$ is varied. A broad unimodal state (at the back) splits into distinct bimodal states (to the front). In the middle a critical point exists, called the cusp (point $P$ in Fig. 3) where the split occurs.

The distributions in Figs 5 show that qualitatively distinct behaviour occurs when $\alpha$ or $\beta$ are varied. For both parameters, a change from a unimodal to a bimodal distribution can be seen. Each distinct shape of the distribution can be identified with one of the potential functions in Fig. 2. In principle, a change in only one of the two structural parameters ($\alpha$ and $\beta$) can move the distribution between unimodal and bimodal forms.

We are now in a position to apply the above to collapse trajectories from climate models.

## 3   AMOC collapse parameter estimation

We describe how to find an optimal solution under the framework described in the previous section. Using a Bayesian optimisation procedure, estimated values of $\beta_0$ and $\delta\beta$ can be found, together with the scaling parameters $\nu$ and $\lambda$. We will also estimate the values for $\mu_{\pm}$, resulting in a six parameter list that describes (the upper branch) of an AMOC collapse.

The parameters $\beta_0$ and $\delta\beta$ are independent of each other, but need to cross the curves $B_{1,2}$ in Fig. 3) to match the corresponding values for $\mu_{\pm}$. This constraint is satisfied by the resulting values for $\alpha_0$ and $\delta\alpha$. (This can still lead to solution candidates

that are not suitable for the collapse trajectories and are eliminated in the sampling process below.) The scaling parameters are not fully independent because $\lambda < \nu$ (the offset cannot exceed the scaling) and knowing where the upper and lower branches are located already gives a rough estimate.

## 3.1 Parameter estimation

Cobb (1978) was able to fit the distribution in Eq. 6 using optimisation techniques (which were numerically unstable and not very flexible). Though the estimates for the scaling parameters $\lambda$ and $\nu$ can be quite good with this approach, estimating the trajectory parameters $\beta_0$ and $\delta\beta$ requires a more flexible method. Knowing which distribution to use, we can estimate the posterior probability distribution of the parameters given the data $\Psi(\mu)$,

$$P(\nu, \lambda, \beta_0, \delta\beta, \mu_\pm \mid \Psi).$$

Bayes' rule tells us the probability of a given observation $\Psi$ given the probability of the parameters (marginal on the left, or posterior) is proportional to the probability given the parameters (marginal on the right, or prior) and the full distribution (likelihood),

$$P(\nu, \lambda, \beta_0, \delta\beta, \mu_\pm \mid \Psi) \propto P(\Psi \mid \nu, \lambda, \beta_0, \delta\beta, \mu_\pm) \cdot P(\nu, \lambda, \beta_0, \delta\beta, \mu_\pm).$$

Sampling different values from the parameters' prior distributions will give corresponding values for the posterior distributions. A Bayesian sampler chooses successive values that tend towards greater likelihood of the model, given the observed trajectory, and will converge towards an optimal fit. Conceptually, this is what an MCMC (Markov chain Monte-Carlo) optimiser does (Bolstad, 2010). A widely used sampling algorithm is the Metropolis algorithm (Hastings, 1970; Bernardo and Smith, 2009), which we also use here. This algorithm has been implemented in many software packages (e.g. Salvatier et al. (2016); Carpenter et al. (2017)).

The sampling process is time consuming because the evaluation of the potential (to calculate $P(\Psi \mid \nu, \lambda, \beta_0, \delta\beta, \mu_\pm)$) requires numerical integration (using a quadrature method), which is costly to evaluate (the exponential family of distributions cannot, in general, be evaluated analytically).

### 3.1.1 Prior distributions

The prior distribution of a parameter represents all the information known about that parameter before confrontation with the observed values (Bolstad, 2010). With $\nu$ and $\lambda$ transform the AMOC state variable ($\Psi$) with a shift ($\lambda$) and a scaling ($\nu$). The shift $\lambda$ cannot exceed the normalisation $\nu$, giving an upper bound on $\lambda$. Also, we note the lower limit of the lower branch, meaning $\lambda$ must be larger than this minimum value. Similarly, the scaling $\nu$ cannot be larger than the maximum value of the AMOC on the upper branch. We expect the linear parametrisation of $\alpha$ and $\beta$ introduced in the previous section to be $\mathcal{O}(1)$.

We are nonetheless still faced with infinite support on the coefficients of the expansion of the parameters ($\beta_0$, $\delta\beta$). We therefore transform $\beta_0$ and $\delta\beta$, with support $(-\infty, \infty)$, using the arctan function to map to $(-\pi/2, \pi/2)$. After such a transformation, we can sample from the flat prior distribution on that interval with most of the probability mass on 'reasonable' values

(i.e. $\mathcal{O}(1)$). The following prior distributions are used:

$$\nu = U(\min(\text{AMOC}), \max(\text{AMOC}))$$

$$\lambda = U(\min(\text{AMOC}), \nu)$$

$$\mu_+ = U(\mu_{\text{S+}}, \mu_{\text{UP}})$$

$$\mu_- = U(\mu_{\text{DN}}, \mu_{\text{S-}})$$

$$\tan(\beta_0) = U(-\pi/2, \pi/2)$$

$$\tan(\delta\beta) = U(-\pi/2, \pi/2),$$

with $\min(\text{AMOC})$ and $\max(\text{AMOC})$ is the minimum/maximum values in an observed collapse trajectory. $U$ is the uniform distribution on indicated intervals. The intervals values of the collapse points $\mu_\pm$ we stipulate as being bounded by where the trajectories merge ($\mu_{\text{UP}}$ and $\mu_{\text{DN}}$) and the inner values ($\mu_{\text{S-}}$ and $\mu_{\text{S+}}$) observed in the trajectories (within which bimodality is demanded, see Fig. 6). [1]

## 3.2 Fitting EMIC collapse trajectories

An AMOC collapse was induced in models of intermediate complexity in Rahmstorf et al. (2005) by applying a freshwater forcing to the North-Atlantic subtropical gyre region that reduced the salinity in the subpolar gyre to its north. Six of these models have a 3-D ocean components; in Fig. 6 the trajectories of those collapses are reproduced (right column, the freshwater flux has been labelled $\mu$ here) together with their numerical derivatives (left columns in the panels). In Tab. 1 the models are listed. The forcing values of $\mu$ are known and the same for each climate model. Each model was run to equilibrium for each forcing value; there is therefore no explicit time dependence in the hysteresis loops shown. Both the AMOC strength and the forcing value have units Sv (=$10^6$ m s$^{-1}$). Note that the bifurcation points ($\mu_\pm$) must lie within the range where the trajectories appear bimodal.

The trajectories are from the numerical Earth System Models (EMICs) Rahmstorf et al. (2005, Fig. 2, bottom panel). The numerical derivatives show where the AMOC changes quickest as a response to the change in freshwater forcing. Each model has two peaks where the changes are largest, one for each change between stable branches. These peaks are located at the repellor in between the two attractors (the stable branches). At the repellor only unstable solutions exist and the AMOC is driven to a stable solution, away from these states.

If no other mechanisms apart from the salt advection are important we expect the bifurcation points to lie beyond the observed transition points because a noise-induced transition pushes the AMOC into the off-state sooner. (Note that although the collapse points are expected to lie before these peaks, low levels of noise will obscure this effect.) The dashed lines indicate the regions where we will search for the optimum values of $\mu_\pm$. These differ from the fixed 0 and 0.2 values chosen by (Rahmstorf et al., 2005), who also shifted the trajectories to align on these values.

---

[1] To exclude parameter values that lead to intersections of $B_{1,2}$ more than once, we artificially decrease the likelihood of these values. The discriminant of the polynomial at each forcing value indicates when this is needed.

| model | ocean component | atmosphere component | reference |
|---|---|---|---|
| Bremen | large-scale geostrophic | energy balance | Prange et al. (2003) |
| ECBilt-CLIO | 3D primitive equations | quasi-geostrophic | Goosse et al. (2001) |
| C-GOLDSTEIN | 3D simplified | energy-moisture balance | Edwards and Marsh (2005) |
| MOM hor | 3D primitive equations (MOM) | simple energy balance | Rahmstorf and Willebrand (1995) |
| MOM iso | as above, with isopycnal mixing | simple energy balance | |
| UVic | 3D primitive equations (MOM) | energy-moisture balance | Weaver et al. (2001) |

**Table 1.** Overview of models used. Each data point is independent from the others because each is the result of a quasi steady state run. The number of data points for each model was regridded onto a uniform freshwater forcing range consisting of 300 points. The summary of the type of model component and references are taken from Rahmstorf et al. (2005).

| model | $\sigma$ | $\mu_-$ | $\mu_+$ | present day |
|---|---|---|---|---|
| Bremen | 0.181 | [-0.018, 0.010] | [0.120, 0.220] | ( 0.070, 18.8)- |
| ECBilt-CLIO | 0.176 | [-0.044, 0.030] | [0.115, 0.210] | (-0.110, 18.2)+ |
| C-GOLDSTEIN | 0.122 | [-0.100, 0.035] | [0.115, 0.190] | (-0.100, 29.0)+ |
| MOM hor | 0.526 | [-0.010, 0.010] | [0.130, 0.200] | ( 0.110, 20.0)- |
| MOM iso | 0.216 | [-0.010, 0.020] | [0.150, 0.210] | ( 0.050, 22.8)- |
| UVic | 0.260 | [-0.020, 0.010] | [0.188, 0.225] | ( 0.080, 25.0)- |

**Table 2.** Overview of models, the estimated standard deviation with the upper branch fitted to a linear function (note that the original trajectories had already been smoothed), the ranges of $\mu_\pm$, the location of present day in the models, and whether the present day value is in the unimodal regime (+) or not (-). All values have units Sv.

Before fitting, the upper and lower branches were extended to the left and right to fill the space of $-0.2 < \mu < 0.4$. A linear fit was use to produce additional values of the corresponding branches (at the same density of those points already present). All models then occupy the same freshwater forcing space. This is desirable because not all models have a lower branch that is fully sampled (specifically, UVic). The lower branch was extended with a negative rate of increase if the lower branch was moving upwards with increasing $\mu$ (MOM hor and MOM iso).

Our main goal is to model the transition from on-branch to the off-branch, that is, the upper right half of the hysteresis curve, and not so much the dynamics that govern the lower branch. Also, because we assume that other dynamics govern the lower branch and, our simple model has to be extended to account for those dynamics. We ignore the data on the lower branch before the collapse point so the fits would not be influenced by these points. We expect the remaining points of the trajectory to be dominated by the salt-advection mechanism.

We start by identifying some characteristic points in the trajectories in Tab. 2. The $\sigma$ (variance of the process) of the models is not given in Rahmstorf et al. (2005) or elsewhere in the literature, but was estimated as the deviation with a fitted function to

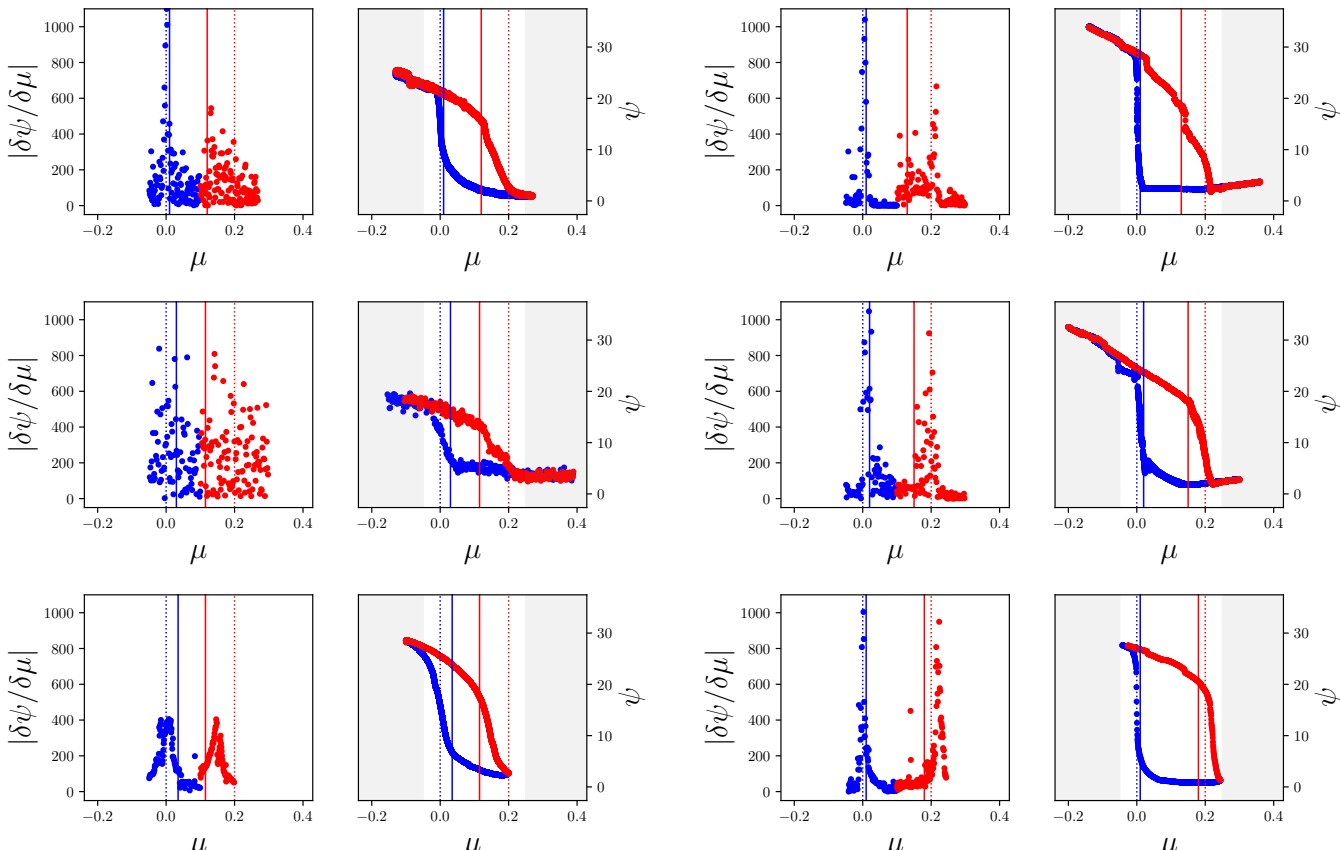

**Figure 6.** Absolute values of numerical derivatives (left) from the trajectories of AMOC strength as function of freshwater forcing to the right (taken from Rahmstorf et al. (2005, Fig. 2, bottom panel), reproduced with permission from the publisher: American Geophysical Union). In red the upper branch, blue the lower branch. Left column: Bremen, ECBilt-CLIO, C-GOLDSTEIN; right column: MOM hor, MOM iso, UVic. Vertical solid lines mark $\mu = 0$ (blue) and $\mu = 0.2$ (red); vertical dashed lines mark the chosen boundary values for $\mu_{\pm}$. All values have units Sv.

the left most the top branch. (Note that smoothing was already applied in Rahmstorf et al. (2005), lowering the variance of the trajectories. Because we want to fit the collapse trajectory as given, we use the variance as evident from the data.) In principle, $\sigma$ could also be estimated as a parameter in the Bayesian optimisation, but that would unnecessarily enlarge the search space. Note that the 'off-state' of the AMOC in these models is not 0, but $\sim 2\text{Sv}$ of AMOC strength. If the salt-advection mechanism were the only operative effect, we expect this value to be $\leq 0$. If a reverse advection cell emerges as the lower hysteresis branch,
this value is negative.

    In Fig. 7 fitted distributions are shown (also tabulated in Tab. 3). As best fit parameters, we choose the mean values of the marginal posterior distributions. The dashed grey line marks the positions of the unstable solution (repellor) in between the two attractor branches which separates the two basins of attraction.

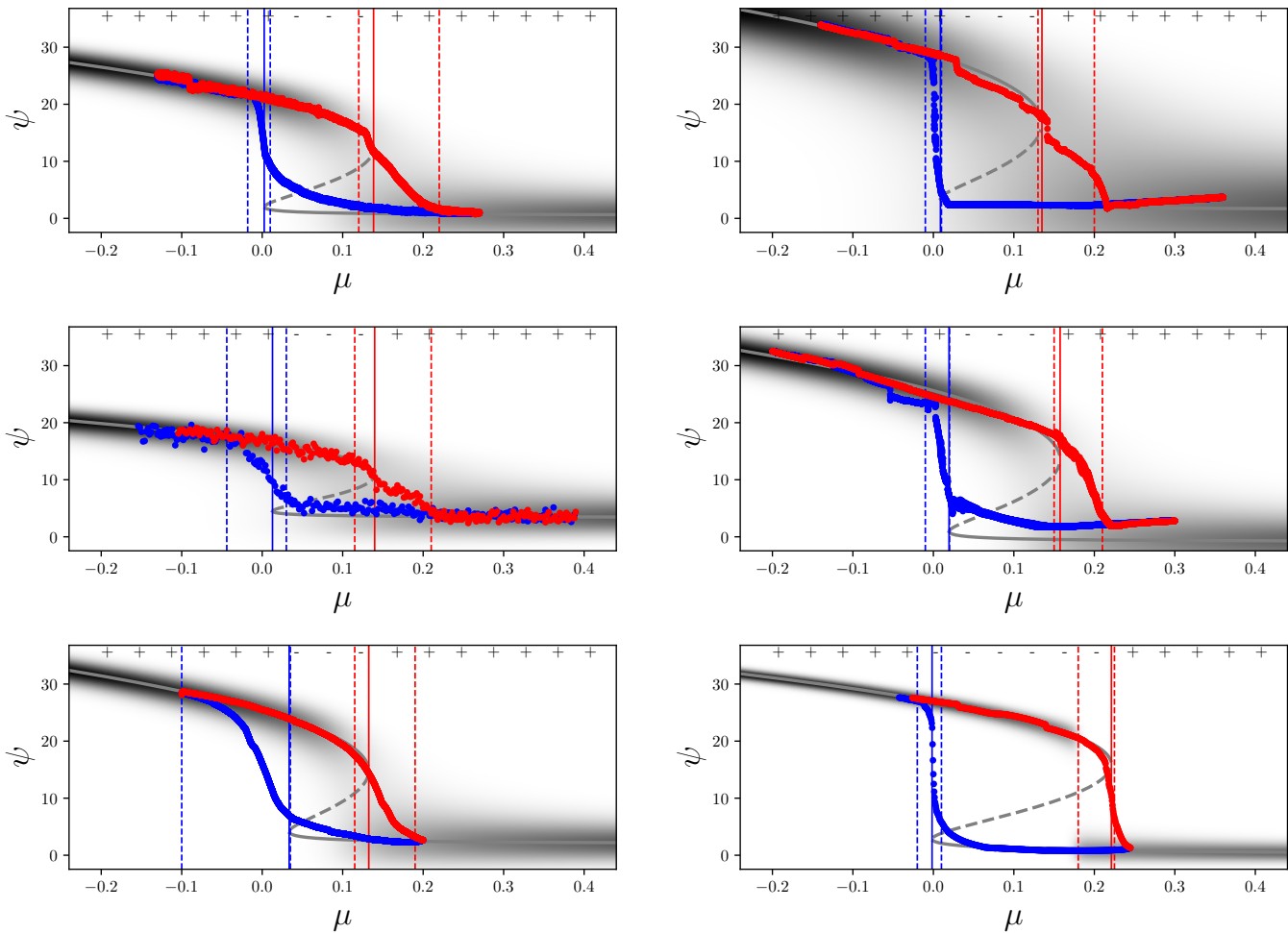

**Figure 7.** Estimated distributions under changing $\mu$. Left column: Bremen, ECBilt-CLIO, C-GOLDSTEIN; right column: MOM hor, MOM iso, UVic. Vertical dashed lines mark the chosen boundary values for $\mu_\pm$, with solid lines the fit values. Grey dashed line indicates the local minimum in the distribution (trench). Top $\pm$ symbols indicate the sign of the discriminant $D$ for the fitted distribution ($+$ for unimodal, $-$ for bimodal). Distribution spreads have been inflated with a factor $\nu/2$ to make them visible. All values have units Sv.

| | $\nu$ | $\lambda$ | $\beta_0$ | $\delta\beta$ | $\mu_-$ [Sv] | $\mu_+$ [Sv] | RMS deviation [Sv] |
|---|---|---|---|---|---|---|---|
| Bremen | 21.2 | 8.44 | 0.28 | -1.32 | 0.002 | 0.14 | 0.38 |
| ECBilt-CLIO | 13.8 | 8.45 | 0.26 | -1.24 | 0.013 | 0.14 | 0.60 |
| C-GOLDSTEIN | 24.2 | 10.7 | 0.27 | -1.39 | 0.033 | 0.13 | 0.48 |
| MOM hor | 28.4 | 11.7 | 0.26 | -1.31 | 0.009 | 0.13 | 0.98 |
| MOM iso | 25.7 | 8.90 | 0.32 | -1.37 | 0.019 | 0.16 | 0.83 |
| UVic | 23.5 | 10.9 | 0.35 | -0.97 | -0.002 | 0.22 | 0.81 |

**Table 3.** Mean values and standard deviations of parameters corresponding to the fitted functions in Fig. 7. The root-mean-square deviation (a goodness of fit measure) has been determined on the upper branch up to the fitted collapse point.

The fits with a linear series through the $(\alpha, \beta)$ parameter space result in a mismatch between the behaviour seen on lower branches and that on the upper branches. This is less obvious for UVic and ECBilt-CLIO, but especially apparent for the two MOM models.

## 4 Discussion and conclusion

We derived a simple model of AMOC collapse based on Langevin dynamics (Eq. 1) with a changing freshwater forcing ($\mu$) and applied this to EMIC simulated collapse trajectories taken from Rahmstorf et al. (2005). The collapse occurs at a bifurcation point $\mu_+$ which appears smaller than given in (Rahmstorf et al., 2005). A corresponding bifurcation point $\mu_-$ relates an abrupt transition back to the on-state. The AMOC also requires an offset and scaling parameter to be fitted ($\lambda$ and $\nu$). These six parameters are sufficient to describe the abrupt collapse of the AMOC as part of a hysteresis loop under varying freshwater forcing.

Any process which allows two stable states with rapid transitions between them and an asymmetric response to the forcing could in principle be described by our method. Other such geophysical processes might be ice sheet mass loss (e.g. Robinson et al. (2012)), forest dieback (e.g. Staal et al. (2016)), and lake turbidity (Scheffer and van Nes, 2007).

The resurgences of the AMOC seen in the hysteresis diagrams behave differently from the collapses. The Langevin model is too simple to capture both processes. It is, however, possible to fit the change in the upper branch of the AMOC–the 'on-state'–as it moves towards a critical point and the dominant salt-advection feedback mechanism breaks down.

We note that Rahmstorf et al. (2005) determine the AMOC strength as the maximum of the meridional volume transport in the North Atlantic and might explain the asymmetry between the two branches. If for a reverse overturning cell the wrong metric has been used then the lower branch location is not correct. It is conceivable that the Langevin model results in better fits if Rahmstorf et al. (2005) had sampled $\max(|\Psi|)$ instead of $\max(\Psi)$, which would have resulted in a better metric of the lower branch. With the metric used it is not apparent whether a reversed overturning cell was present or not because it was not sampled if the AMOC had taken on a negative value. It is unclear to what extent the models discussed here develop a reversed

overturning circulation which can arise in 3D models (Weijer and Dijkstra, 2001; Yin and Stouffer, 2007), but which can also be suppressed by atmospheric feedbacks (Yin and Stouffer (2007); however, see also Mecking et al. (2016)), and strongly affected by gyre dynamics (Prange et al., 2003). These effect are not captured by the simple Langevin model proposed here, but at present it is still unclear to what extent these effects are essential in capturing the first order stability properties of the AMOC. In each case, there is no obvious way to model the asymmetry between the two branches, and obtain a full description. The two branches could be separated by associating each with a different overturning cell. The upper branch is identified with the NADW-driven cell, while a reverse cell is responsible for the lower branch. If indeed a reverse overturning cell (as described in e.g. Yin and Stouffer (2007)) dominates the lower AMOC branch, two separate overturning cells are responsible for the observed trajectories, and the two branches then cannot be expected to fit with the same parameter set.

However, another possible explanation is that (two) separate mechanisms are responsible for the upper and lower branch dependency on $\mu$. Possible mechanisms include possible mechanisms include the influence of wind-stress, North Atlantic subpolar gyre convective instability (Hofmann and Rahmstorf, 2009), or other pathways of deep water formation (Heuzé, 2017). Also, changes in the ITCZ (inter-tropical convergence zone) due to ocean-atmosphere feedbacks are possible (Green et al., 2019); these can, in turn, can affect the salinity of the North Atlantic subtropical gyre region. However, Mecking et al. (2017) showed that for a high-resolution model the salt-advection feedback was nevertheless stronger than the ITCZ effects. Other wind coupling can occur further south through a coupling with the ACC (Antarctic Circumpolar Current) which is based on the thermal wind relation (Marshall and Johnson, 2017).

A third explanation is that deep water formation is a local process, and as a result an asymmetry is to be expected between the two branches. Local convection can, however, be subject to global controls and be associated with a sinking branch which occurs in conjunction with deep convection, but is not directly driven by it, see Spall and Pickart (2001) for a detailed discussion. The AMOC could develop a reverse cell where the overturning is driven by Antarctic Intermediate Water (AAIW), which is not part of the conceptual picture presented here (Yin and Stouffer, 2007; Jackson et al., 2017). The reverse cell introduces an asymmetry in the collapse trajectories because the driver of deep water formation is not in the North Atlantic, and might break our assumption that both the on and off branches are controlled by the same process. It is therefore difficult to estimate the return path of the AMOC if the lower branch has additional drivers from the dominant salt-advection mechanism of the upper branch. Forcing values appropriate for the lower branch might be different than those found for the upper branch.

Furthermore, the methodology used in this paper comes with difficulties in the numerical implementation. The fit procedure requires the normalisation of each distribution in the $\mu$ timeseries. Because no analytic solution exist a numerical approach is needed. The numerical integration adds to the computational costs of the fits. The Markov chain method is also prone to find local optima. Also, the cost of numerical integration necessitates stopping the fits at shorter chains than (perhaps) are needed, an analytic formulation of the integrand would alleviate this but none exists to our knowledge. Modern sampling algorithms allow for gradient information to be used, which is effective when sampling a higher dimensional parameter space (the Metropolis algorithm used in this paper has greater difficulty as the dimensionality of the parameter space increases). Tighter constraints on the prior distributions could be beneficial here.

As stated in Rahmstorf et al. (2005), the EMIC trajectories had already been smoothed, resulting in a smaller variance; a smaller variance leads to distributions that are more sharply peaked. This increases the computational cost of integrating the distributions numerically. Smoothing can also add to the inertia seen in the collapses, but might be due to other reasons such as stopping the EMIC simulations before equilibration of the AMOC collapse, leaving the AMOC in a winding-down state. Also, the models in Rahmstorf et al. (2005) were integrated for 1000 model years per freshwater forcing value (which was changed

in 0.05 Sv increments). If the integrations were done for an insufficient amount of time, the AMOC collapse is incomplete, leaving the measured value out of equilibrium. The intermediate points in the collapse trajectories beyond the bifurcation points indicate that either the sample points are inaccurate or other processes are involved in the AMOC.

    Finally, the fitted collapse trajectories were done on an ensemble of EMICs, which arguably are not sufficiently representative of the real climate. As noted by Gent (2018), the hysteresis behaviour has not been investigated fully in models of greater

complexity than EMICs; the computational cost being the prohibitive for models with high resolution (and short time steps). The hysteresis behaviour in glacial state changes has, however, been investigated in greater detail using models with simplified dynamics (e.g. Schiller et al. (1997); Zhang et al. (2017)). The question arises to what extent the procedure outlined in this paper can be applied to more complicated models such as those in the CMIP archives (Taylor et al., 2012). These models do not show a full collapse trajectory like those in Rahmstorf et al. (2005), which means no sample points of the lower branch are available.

Also, CMIP provides times series of forced runs. To validate our method, a transient run requires known equilibrium bifurcation points, under a slowly changing $\mu$, and include an AMOC collapse. Using a simple box model, transition probabilities for an AMOC collapse have been determined by Castellana et al. (2019). From the CMIP ensemble a similar estimate might be obtained, or at least the collapse characteristics of various models can be compared. Provided the CMIP models accurately capture the behaviour of the real AMOC and the freshwater forcing counterpart (our $\mu$) can be identified, an estimate can be

made of the distance of the current climate state to the collapse point. Freshwater quantities such as $M_{ov}$ have been posited (e.g. Drijfhout et al. (2011)) as being suitable indicators of AMOC stability. It is possible that $M_{ov}$ relates to $\mu$ and can be used to extend our method to transient runs, but at present it is unknown whether this can be done. The inclusion of icesheets can make a substantial difference in AMOC recovery (Ackermann et al., 2020). Also, the atmospheric freshwater transport might have a stabilising effect on the AMOC that is greater than the freshwater transports by the ocean (Lohmann, 2003). There is, however,

also evidence that coupled climate models suffer from a salinity bias that favours an AMOC that is too stable (Drijfhout et al., 2011; Liu et al., 2017). These matters are outside the conceptual picture of $M_{ov}$ as a stability indicator. It is therefore still an open question how probable an AMOC collapse is in more realistic models, and reality, but with the method outlined in this paper a first step could be made in answering this question.

*Author contributions.* S.D conceived the original idea. J.B. developed the theoretical formalism, performed the calculations, and prepared

the figures. J.B., S.D., and W.H. contributed to the final version of the manuscript.

*Competing interests.* The authors declare that they have no conflict of interest.

*Acknowledgements.* This work was partially funded by the European Commission's 7th Framework Programme, under Grant Agreement number 282672, EMBRACE project. The authors thank the two anonymous referees and editor Prof. Dr. Lohmann for their valuable comments and suggestions that have improved the manuscript greatly. The authors also thank Prof. Dr. Rahmstorf for providing the original data.

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
