# Peer review of "Characterisation of Atlantic Meridional Overturning hysteresis using Langevin dynamics"

_Earth System Dynamics, 2020_

## Referee Comment (RC1) · Anonymous Referee #1 · 3 Aug 2020

The paper by van den Berk et al "Collapse of the Atlantic Meridional Overturning described by Langevin dynamics" is an interesting application of the classic analytical approach of Poston and Stewart with introduced stochasticity for modelling AMOC trajectories of the EMICs published in [Rahmstorf et al 2005]. I think the paper should be published after a minor revision.

The title should be corrected: "Modelling collapse of the Atlantic Meridional Overturning using the Langevin dynamics". As the authors admit themselves, EMICs are not sufficiently,representative of the real climate. Also, given the number of parameters the authors use to fit their model (six) and their geometrical origin (see description of $\nu$ and $\lambda$), I understand why the authors claim that only the freshwater forcing is the variable that determines the dynamical behav

It would be interesting to see how the model can be used for forecast of bifurcations.

[Figure]

The authors perform derivation of the model parameters using Bayesian framework, but once the model has been fully formed and the parameters are obtained for several EMICs, can the authors attempt forecast or hindcast of the bifurcating time series?

[Rahmstorf et al 2005] paper used 11 models and only hysteresis loops were presented (not actual AMOC trajectories)

https://agupubs.onlinelibrary.wiley.com/doi/pdfdirect/10.1029/2005GL023655?download=true

Can a figure be added with plotted time series that could be derived from the obtained model? For example, for the set of parameters averaged over a set of the selected EMICs? I wonder how realistic could be the time series and at what time scale it could forecast an AMOC bifurcation?

I understand that the framework is quite heavy computationally. Can the authors add discussion on how applicable can be this approach in other areas of geosciences where similar potential models may be used?

The authors derived datasets from the published figures - is it allowed practice? Shouldn't they be obtained from the authors as datasets? Can the authors add information about the derived datasets in the table (number of points, etc)? Also, can more recent EMICs be used?

Further comments

The abstract should be modified to say that model is fitted to the trajectories.

In the first paragraph, AMOC acronym is introduced twice.

Instead of "invigoration" it is better to say "re-activation".

Line 90 - "diagrams"

Figure 2 - labels in all panels should be of the same font size

Line 124 - "the simplest"

[Figure]

Line 152 - grey lines are mentioned in Figure 4, not clear which, maybe make them dashed? Similarly, dashed lines in Figs. 6,7 are impossible to see - enlarge these figures and all labels.

Table 1 should be expanded to include more information on the selected models - countries, resolution, etc.

---

## Referee Comment (RC2) · Anonymous Referee #2 · 6 Aug 2020

This is an interesting paper, which aims to describe the collapse and hysteresis of the AMOC observed in intermediate complexity climate models subject to freshwater forcing by low-dimensional Langevin dynamics as a stochastic bifurcation of a double well potential. Substantial revisions are necessary to improve the clarity of the manuscript and to support the conclusions.

General comments

1.) It is not clear what the purpose of the paper is. The authors do not state what their model is able to explain or predict. Is the purpose to predict the exact parameter value of a collapse? Or at least to develop a method to do this? Are there prospects to apply the method to observational data? Or is an aim to understand dynamically what is happening in realistic climate models? This should be stated in the introduction

(P2L29ff). It is also unclear whether they want to only/mostly model the AMOC collapse (as stated at some points in the paper) or also the resurgence.

2.) Regarding the conclusions, how can the authors say that the model successfully captures the dynamics? They don't compare with other models of higher or lower complexity, nor do they have any metric that shows goodness of fit or anything similar. This would be necessary to make such a conclusion.

3.) The manuscript is not very well written and hard to follow. The terminology is often unclear. (E.g. what is a "track", and how does the use of "stability landscape" apply here? See specific comments.) Some corrections are given under "technical corrections", but the language and terminology has to be generally improved throughout the manuscript. Furthermore, I believe the manuscript can be shortened severely. What the authors want to get across can be said more efficiently. Many things are mentioned twice or more (see specific and technical comments for suggestions). Finally, the labels in multiple figures are unreadable.

4.) The data acquisition seems problematic. I am not sure whether it is viable for this journal to present a data analysis based on visually extracted data from a figure of another publication. Accordingly, the quality of the data is a major drawback of the study (e.g. arbitrary smoothing and AMOC metric). Their main problem in fitting the data might be due to the specific metric that is shown in the Rahmstorf et al. (2005) figures, so it is a shame that the authors are not able to resolve that.

5.) The description of their method contains many errors, and is incomplete. An explicit expression for the likelihood, as well as details of the Metropolis-Hastings implementation are missing. In the discussion, the authors name difficulties in the numerical implementation as a possible reason for the failure of their fit to describe the lower AMOC branch, but it is for the reader not possible to assess whether this is relevant, since no details or robustness tests are given. Furthermore, it is not stated how many data points the respective data sets contain, and it is not mentioned that the authors

assume successive data points to be independent. It is also not mentioned how the maximum of the posterior parameter distributions is picked.

6.) Finally, several questions regarding the methodology. a) Why do the authors not try to estimate sigma with their Bayesian method? Why not include observational noise? This could handle the fact that the data is filtered arbitrarily. It could also completely change the locations of the inferred bifurcation points. b) To make the paper more understandable it would be good to note explicitly early in the manuscript that the movement of mu is actually known. c) Why not try multiplicative noise? (see also e.g. Das/Kantz Phys. Rev. E 101, 062145, 2020) This should relatively easily give a model that describes the asymmetric behavior. d) It should be noted explicitly that there is no time dependency of the data. I wonder why they choose not to fit to time series instead? This would allow to treat the non-equilibrium nature of the data. Also, it would be much more applicable to observational data and to make predictions. e) Why not only move along alpha at a certain fixed beta? Is moving both parameters supported by the data significantly better?

Specific comments

Abstract: "Machine learning": To my knowledge MCMC is not considered a machine learning technique. The abstract needs to be expanded to better reflect the motivation of the study, what their method enables them to do, and their conclusions.

P2L42-45: This is a not a very clear explanation of the salt advection feedback. The main point is that North Atlantic salinity anomalies (positive/negative) are amplified by their effect on the overturning flow (strengthening/weakening), the strength of which controls the North Atlantic salinity. This is thus a positive feedback and leads to bi-stability with the associated possibility of abrupt transitions.

P2L53: "...number of solutions for a given value of the freshwater forcing goes from 2 to 3...". Should say "goes from 3 to 1" as the bifurcation point is crossed. (There are 2 solutions precisely at the bifurcation point, but I think this saddle-node fixed point is

not relevant here.)

P3 Caption Fig.1: The terminology of this figure is not appropriate and furthermore not understandable at this point within the manuscript. No trajectory is shown, but a bifurcation diagram. They have to be more specific with what they mean by a deformation of the "trajectory". Also, at this position within the manuscript, it is completely unclear what they mean with "trench of the distribution". Either leave out or explain in the main text. Furthermore, I suggest to use the term "resurgence point" for mu-, and use that terminology throughout the paper. Note that e.g. in P5L91, mu+/- are being referred to as "collapse points".

P4L64: Can the authors elaborate why they think a double well potential has mainly been studied qualitatively? I would argue that this simple and general mathematical model has been studied quantitatively to an exceptional degree.

P4L65ff: It is a bit confusing when the authors first say that 2 parameters are enough to describe bi-stability, but then use another 2 parameters to scale and shift to the AMOC variable. Maybe it would be better to first explicitly say that by a shift and scale of the variable x, one can eliminate the third order term as well as the fourth order coefficient. Both of these transformation do not influence the global bifurcation behavior. Then, they can state that a shift and scaling is considered when fitting to the climate model data.

P4L78-81: Can the authors elaborate why they obtain these rough estimates for the parameters, and how they are insensitive to other parameter values?

P5L90-91: When speaking about "solution" what exactly do the authors mean?

P5L92-97: This section is a bit unclear. Can the authors define a "track", and what does it mean to be one-dimensional? The fact that alpha and beta are called normal and splitting factor is better mentioned earlier. A more clear distinction of "parameter" and "variable" would be appropriate.

P5L101: This argument is unclear to me. The fact that the AMOC is scalar variable should not constrain the path through the stability landscape in any way. Do the authors rather want to say that in the climate model experiments there is only a single control parameter mu, and that by assuming a linear dependency of both alpha and beta on mu, they can express some parameters by the extremal values of mu?

P8L127-129: Maybe the authors can elaborate more specifically on why these arguments are relevant in order to neglect a non-linear change of either mu or alpha/beta?

P8L141-146: Improve this explanation. When introducing stochasticity, the asymptotic dynamics for each parameter value give rise to a stationary density. In the case of the scalar potential, this distribution can be given analytically up to a normalization factor. Thus, the distribution can be used as a likelihood function (if I understand correctly) for parameter inference with MCMC.

P9L157: What is the "sampled" bi-modality region?

P9L160: "These changes correspond directly to the potential functions in Fig. 2." What is meant by this?

P10L174: Can the authors explain why lambda < nu in general?

Caption Figure 5: The terminology is unclear. What is meant by a "singular" maximum? What is meant by the dominant and the weak mode, and what is the inversion? There are also grammatical errors ("...in the middle and inversion from weak...").

P11L180: I think a more precise statement would be that they estimate the posterior probability distribution of the parameters, given the data psi(mu). Furthermore, the following equation does not define the likelihood but the posterior distribution.

P11L183: Why linearised?

P11L184-188: This statement of Bayes' rule is not correct, please revise. The right hand side is not called Bayes' factor (which arises in model comparison).
P12L209-219: The constants muS+ etc. are not properly introduced and should be shown in one of the figures. The footnote 2 needs to be explained better.

Caption Table 1: Why is a linear function used and not a higher order polynomial? This does not seem to be very suitable to the data.

P13L228-230: This is not very precise wording. What do the authors mean with "unstable" and "more stable" solutions?

P14L242-245: What do the authors want to say here? It comes as a surprise to me that suddenly only the data for mu > mu+ should be relevant? And why do they now claim that the model C-GOLDSTEIN does not "appear" to show a collapse?

P14L252-253: Unclear what the authors are trying to say.

P14L259: What is meant by "non-linear degradation"?

P15L267: In what way is the model sufficient to describe the data? Certainly the "re-invigouration" is not well captured.

P17L308-310: Unclear what is meant here. What are "non-admissible" solutions?

P17L313-315: Unclear. Smoothing might be due to other reasons?

P17L323: What is meant by "direct numerical stochastic integration"?

P17L330: How exactly does this paper present a step forward to assess the likelihood of a future collapse of the AMOC? The method presented here relies on previously modeled collapses of the AMOC with realistic climate models. How does the method generate additional information?

Technical corrections

P1L15: last glacial maximum and early holocene -> last glacial period

P2L26: . . . which are presumed to be functions . . .

[Figure]

P2L30: ...and tipping points in the climate, it has not been . . .

P2L38: . . . or an increased surface freshwater flux by changes in precipitation minus evaporation. . .

P2L46: scalar variable obtained by integrating ...

P2L52: . . . one of the two basins of attraction vanishes . . .

P7L126: rather "(delta alpha, delta beta)"?

P8L142: Remove: As shown by Cobb (1978), this distribution belongs to the exponential family.

P8L144: The polynomial potential introduced in the previous section, we had already. . ., gives the probability . . .

P8L148: Note that C = C(alpha,beta), which does not have...

P8L150: . . . because of the scaling ...

P8L152 and Fig. 4 caption: In what way is this a sample collapse trajectory, or an example trajectory?

P9L161: . . . a change in only one . . .

P9L163-165: Why not say this at P8L148? It is a bit redundant otherwise.

P10L168: arrived at -> described

P10L171: independent of each other

P11L180: The method used is not considered machine learning.

P11L190: Does not seem to be relevant, as it is not done here.

P11L194-196: This is partly redundant, and it is not clear why the authors mean that the model can be fit with uninformative priors.

P11L199-200: Redundant.

P11L207: Redundant.

P11L207-208: An overview of priors is: The following prior distributions are used:

P14L243: do -> to

P14L254: Why "sample" paths? The authors are showing distributions, which is exactly contrary to showing sample paths.

P14L255: Redundant.

P15L265: couples -> models. Why "additionally"?

P16L275: Do they mean arg(max(|psi|)) ?

P17L309 till its -> until it

---

## Author Comment (AC1) · 8 Sep 2020

The paper by van den Berk et al "Collapse of the Atlantic Meridional Overturning described by Langevin dynamics" is an interesting application of the classic analytical approach of Poston and Stewart with introduced stochasticity for modelling AMOC trajectories of the EMICs published in [Rahmstorf et al 2005]. I think the paper should be published after a minor revision.

R: We thank the reviewer for helpful comment and suggestions. Below our responses.

The title should be corrected: "Modelling collapse of the Atlantic Meridional Overturning using the Langevin dynamics".

R: Our suggestion would be "Characterisation of Atlantic Meridional Overturning hys-

teresis loop using Langevin dynamics" to emphasise the purpose of the paper better: using a reduced set of numbers to quantitatively describe the AMOC collapse under a freshwater forcing.

As the authors admit themselves, EMICs are not sufficiently,representative of the real climate. Also, given the number of parameters the authors use to fit their model (six) and their geometrical origin (see description of $\nu$ and $\lambda$), Iunderstandwhytheauthorsclaimthatonlythef reshwaterf orcingisthevariablethatdeterminesthedynamicalb

R: There seems to be a typesetting problem here that renders some of the comment unreadable. However, the other possible forcing effect is thermal, and in principle a sufficiently large warming could also halt deep water formation and collapse the AMOC. Since temperature changes are not a forcing variable in the climate models, we have not included this effect.

It would be interesting to see how the model can be used for forecast of bifurcations. The authors perform derivation of the model parameters using Bayesian framework, but once the model has been fully formed and the parameters are obtained for several EMICs, can the authors attempt forecast or hindcast of the bifurcating time series?

R: We thank the reviewer for this interesting comment which could be explored in further research. A forecast from a partial AMOC weakening series would require a continuation of the freshwater forcing timeseries, and maybe making use of the EMIC derived values as estimates. The value of te freshwater forcing is crucial, however, but this may be estimated from climate projections

[Rahmstorf et al 2005] paper used 11 models and only hysteresis loops were presented (not actual AMOC trajectories) https://agupubs.onlinelibrary.wiley.com/doi/pdfdirect/10.1029/2005GL023655?download=true

R: This is a link to Rahmstorf &a (2005) and another set of even simpler models (with a total of 11 models) are studied in that paper. Those models only have an energy

balance model as the atmospheric component. We ignored those because of their simplicity and argue that their characterisation is too far removed from the real world or CMIP class models.

Can a figure be added with plotted time series that could be derived from the obtained model? For example, for the set of parameters averaged over a set of the selected EMICs? I wonder how realistic could be the time series and at what time scale it could forecast an AMOC bifurcation?

R: This is a good suggestion, but unfortunately no timeseries were given in the published data. To derive those new runs have to be made. The hysteresis loops are obtained by changing the forcing with small steps and then obtaining a new (quasi) equilibrium state for the changed forcing.

I understand that the framework is quite heavy computationally. Can the authors add discussion on how applicable can be this approach in other areas of geosciences where similar potential models may be used?

R: In principle, any hysteresis curve that is produced under a forcing where the lambda and nu transformations suffice to normalise the curve could be used. Other geophysical processes might be icesheet mass loss (e.g. Robinson, Alexander & Calov, Reinhard & Ganopolski, Andrey. (2012). Multistability and critical thresholds of the Greenland Ice Sheet. Nature Climate Change. 2.), forest dieback (e.g. Staal, A., Dekker, S.C., Xu, C. et al. Bistability, Spatial Interaction, and the Distribution of Tropical Forests and Savannas. Ecosystems 19, 1080–1091 (2016)), or lake turbidity (Scheffer, M., van Nes, E.H. Shallow lakes theory revisited: various alternative regimes driven by climate, nutrients, depth and lake size. Hydrobiologia 584, 455–466 (2007). Any process which allows two stable states with rapid transitions between them and an asymmetric response to the forcing could be described by our method. We will add the above as an additional paragraph to the discussion (with the above references included not inline, but as usual).

The authors derived datasets from the published figures - is it allowed practice? Shouldn't they be obtained from the authors as datasets? Can the authors add information about the derived datasets in the table (number of points, etc)? Also, can more recent EMICs be used?

R: We strongly support the development of open science and making data findable and accessible. Unfortunately we have been unable to obtain the original datasets from the authors (we unfortunately received no reply to our request). The (individual points of the) measured values were retrieved from the plots by inverting the transformation matrices. This can be done for certain plots that are converted to pdf from plotting software such as Matlab. The dataset is then numerically the same as the set used to produce the plots in Rahmstorf &a (2005). The publisher allows for the use of individual graphics from their publications: we will note this in the text.

We will include a table with additional information about the dataset.

In principle any hysteresis curve can be used, but we have not expanded the data set beyond the Rahmstorf &a set.

Further comments

The abstract should be modified to say that model is fitted to the trajectories.

R: To be corrected.

In the first paragraph, AMOC acronym is introduced twice.

R: We will remove the 2nd mention.

Instead of "invigoration" it is better to say "re-activation".

R: To be corrected with "resurgence" as suggested by our other reviewer.

Line 90 – "diagrams"

R: To be corrected.

Figure 2 - labels in all panels should be of the same font size

R: We will replot with the same label size.

Line 124 - "the simplest"

R: To be corrected.

Line 152 - grey lines are mentioned in Figure 4, not clear which, maybe make them dashed? Similarly, dashed lines in Figs. 6,7 are impossible to see - enlarge these figures and all labels.

R: We will replot with a colour different from grey and enlarge the labels.

Table 1 should be expanded to include more information on the selected models - countries, resolution, etc.

R: We will include such a table.

---

## Author Comment (AC2) · 8 Sep 2020

This is an interesting paper, which aims to describe the collapse and hysteresis of the AMOC observed in intermediate complexity climate models subject to freshwater forcing by low-dimensional Langevin dynamics as a stochastic bifurcation of a double well potential. Substantial revisions are necessary to improve the clarity of the manuscript and to support the conclusions.

R: We thank the reviewer for the detailed comments and valuable suggestions. Below are our follow up comments and actions.

General comments 1.) It is not clear what the purpose of the paper is. The authors do not state what their model is able to explain or predict. Is the purpose to predict the

exact parameter value of a collapse? Or at least to develop a method to do this?

R: Our aim is to investigate whether it is possible to model the outcome of complex models with a low dimensional model and thereby enhance understanding of the physics of an AMOC collapse and its hysteresis behaviour. The Langevin model defines a low-dimensional manifold that captures the essential collapse characteristics. To the extent that the low-dimensional model is successful in capturing the more complex model this investigation can indeed be seen as the development of a method to predict the parameter range where in a model a collapse would occur. The method is thus partly geared toward providing a means of prediction, but (at present) mainly to provide some characterisation of the collapse that will allow comparison between climate models. If a good fit can be found, then we can further explain the non-dynamical nature of the AMOC variations. As we show, this is partially the case.

Are there prospects to apply the method to observational data?

R: Since the forcing is a freshwater anomaly in the North Atlantic, we would need to estimate the counterpart in the real world. Moreover, the forcing values at the bifurcations have to be known. At present, this is not the case for the real world, and in models it is model-dependent. An attempt could be made to relate the forcing to an indicator that can be linked to the bifurcation points, for instance Mov. This should first be tested. If this can be done, then from the transient change in mu and with knowledge of sigma from observed AMOC variations, a predictive model for the likelihood of an AMOC collapse could be developed.

Or is an aim to understand dynamically what is happening in realistic climate models? This should be stated in the introduction

R: We would like to see this paper as a first step to say more about the behaviour of complex climate models. In particular, we propose a simplified low-dimensional model that is able to explain (and predict) bifurcation (tipping) points and abrupt change in the AMOC. Ultimately, this could be used to investigate abrupt behaviour in CMIP

models, and how likely abrupt changes would be, even if not simulated by the model. Without a-priori knowledge of the freshwater-forcing values associated with the model's bifurcation points, first must be investigated whether mu can be linked to a general indicator like Mov (see the comment above). We will state the purpose of the paper more clearly in the introduction and add the outlook above to the discussion section.

(P2L29ff). It is also unclear whether they want to only/mostly model the AMOC collapse (as stated at some points in the paper) or also the resurgence.

R: We are mainly interested in modelling the collapse, not the resurgence. We will note this here as "Although the hysteresis loops of the AMOC include both a collapse and a resurgence at, we will only attempt to model the collapse from the the stable on branch to the stable off branch."

2.) Regarding the conclusions, how can the authors say that the model successfully captures the dynamics? They don't compare with other models of higher or lower complexity, nor do they have any metric that shows goodness of fit or anything similar. This would be necessary to make such a conclusion.

R: We thank the reviewer for this comment. We will add the posterior spread to show the goodness of fit. To explore and compare with other low-dimensional models than the Langevin model is beyond the scope of this paper (the six included already form a multi-model ensemble). A possible next step could be to apply the Langevin model to a transient run where mu depends on time. The AMOC should eventually collapse and we can have greater confidence in the applicability of our our approach.

3.) The manuscript is not very well written and hard to follow. The terminology is often unclear. (E.g. what is a "track", and how does the use of "stability landscape" apply here? See specific comments.) Some corrections are given under "technical corrections", but the language and terminology has to be generally improved throughout the manuscript. Furthermore, I believe the manuscript can be shortened severely. What the authors want to get across can be said more efficiently. Many things are mentioned

twice or more (see specific and technical comments for suggestions). Finally, the labels in multiple figures are unreadable.

R: We critically reviewed the text. We comment on identified issues with the specific comments below. In short, the terminology will be clarified, and parts of the text will be removed. The figures will be improved with larger and consistent fonts for the labels.

4.) The data acquisition seems problematic. I am not sure whether it is viable for this journal to present a data analysis based on visually extracted data from a figure of another publication. Accordingly, the quality of the data is a major drawback of the study (e.g. arbitrary smoothing and AMOC metric). Their main problem in fitting the data might be due to the specific metric that is shown in the Rahmstorf et al. (2005) figures, so it is a shame that the authors are not able to resolve that.

R: Indeed, the data has been obtained from the figures of the paper. However, it is not visually extracted as the reviewer suggests. The figure we used is a vector graphic and the dataset can be retrieved from it by inverting the plot matrices used to map the original data to the values in the graph. We can replicate the data in this manner. In order to validate this method, we asked for the data from Rahmstorf et al. (2005) but have not received a reply at present. Less smoothing, and presumably larger noise levels, would likely show a stochastic collapse more easily. We will elaborate on the data acquisition in the paper but want to emphasize that the main goals is to develop and test a method to capture complex model behaviour with a simple low-order model. The methodology described here is not affected by smoothing, only the assessment how well the method works is somewhat hindered by this.

5.) The description of their method contains many errors, and is incomplete. An explicit expression for the likelihood, as well as details of the Metropolis-Hastings implementation are missing.

R: We agree that more detail could be given, but we believe a description of the Metropolis-Hastings algorithm is too much detail for this paper as it is well established

and already described in many textbooks. We will add a reference to the textbook of Bernardo & Smith which described it.

In the discussion, the authors name difficulties in the numerical implementation as a possible reason for the failure of their fit to describe the lower AMOC branch, but it is for the reader not possible to assess whether this is relevant, since no details or robustness tests are given. Furthermore, it is not stated how many data points the respective data sets contain, and it is not mentioned that the authors assume successive data points to be independent.

R: Indeed, it is important to assess the robustness of our implementation. To further detail the validity methodology and outcome, we will add a table with model characteristics and state that each point is independent.

It is also not mentioned how the maximum of the posterior parameter distributions is picked.

R: The most likely value is the mean of the posterior distribution. This does assume that the posterior distributions are unimodal. We will discuss this in the text.

6.) Finally, several questions regarding the methodology. a) Why do the authors not try to estimate sigma with their Bayesian method?

R: In principle this can be done. The variation in the hysteresis loops appears constant and can therefore be estimated more easily by other means. This does add to the computational costs and expands the search space, making it more difficult to find solutions. Therefore, we did not follow this method. We will mention this in the text. Why not include observational noise?

R: Observational noise of the real AMOC would have a larger spread. Synthetic series on the basis of the found parameters could be generated with such a noise level. However, we intended to fit the intermediate complexity model outcome, using the data published. AMOC collapse and its likelihood at a given point in parameter space is

model-dependent. One of the essential parameters in this, is the model-dependent sigma. Therefore, we preferred to estimate the sigma of the particular model.

This could handle the fact that the data is filtered arbitrarily. It could also completely change the locations of the inferred bifurcation points.

R: The bifurcation points are determined by the limit (non-stochastic) solutions. A noise driven transition could occur, however, and push the points that bound the hysteresis curve further inwards, towards each other. We will mention this limitation more clearly in the paper.

b) To make the paper more understandable it would be good to note explicitly early in the manuscript that the movement of mu is actually known.

R: Agreed, we will add "The forcing values of mu are known and the same for each climate model." However, the values of mu+ and mu- are not, and model-dependent.

c) Why not try multiplicative noise? (see also e.g. Das/Kantz Phys. Rev. E 101, 062145, 2020) This should relatively easily give a model that describes the asymmetric behavior.

R: Multiplicative noise is state dependent, while we have made the assumption that the noise level is constant; therefore, we used additive noise. We will discuss this point in the text.

d) It should be noted explicitly that there is no time dependency of the data. I wonder why they choose not to fit to time series instead? This would allow to treat the non-equilibrium nature of the data. Also, it would be much more applicable to observational data and to make predictions.

R: Indeed, we will add a remark that each data point corresponds to a fixed value of mu and is not time dependent. Each mu value represents a separate climate model run which has run (more or less) to equilibrium. We need to be able to estimate mu+ and mu- from the models, and this can only be done from a hysteresis-curve which indeed

contains equilibrium solutions and no time-dependence. We agree that a logical next step is to apply the method on time-dependent runs, but to validate the model it is needed that we know the equilibrium bifurcations points as well for that model, and that the time-dependent runs are based on a slowly-changing mu and include an AMOC collapse. There are not many models available that answer all these requirements.

e) Why not only move along alpha at a certain fixed beta? Is moving both parameters supported by the data significantly better?

R: Early attempts with a fixed beta resulted in worse fits; therefore, we opted not to restrict beta. This will be mentioned in the revised text.

Specific comments Abstract: "Machine learning": To my knowledge MCMC is not considered a machine learning technique. The abstract needs to be expanded to better reflect the motivation of the study, what their method enables them to do, and their conclusions.

R: We will remove mentioning machine learning. We will also add to the abstract: "The Langevin model allows for comparison between models that display an AMOC collapse. Variation between climate models studied here is mainly in the strength of the stable AMOC and the strength of the response to a freshwater forcing."

P2L42-45: This is a not a very clear explanation of the salt advection feedback. The main point is that North Atlantic salinity anomalies (positive/negative) are amplified by their effect on the overturning flow (strengthening/weakening), the strength of which controls the North Atlantic salinity. This is thus a positive feedback and leads to bi-stability with the associated possibility of abrupt transitions.

R: We will rewrite as suggested.

P2L53: ". . .number of solutions for a given value of the freshwater forcing goes from 2 to 3. . . ". Should say "goes from 3 to 1" as the bifurcation point is crossed. (There are 2 solutions precisely at the bifurcation point, but I think this saddle-node fixed point is

not relevant here.)

R: We will correct this as suggested.

P3 Caption Fig.1: The terminology of this figure is not appropriate and furthermore not understandable at this point within the manuscript. No trajectory is shown, but a bifurcation diagram.

R: Correct, we will rename to "bifurcation diagram".

They have to be more specific with what they mean by a deformation of the "trajectory". Also, at this position within the manuscript, it is completely unclear what they mean with "trench of the distribution". Either leave out or explain in the main text.

R: We will remove the use of "trench" from the text.

Furthermore, I suggest to use the term "resurgence point" for mu-, and use that terminology throughout the paper.

R: Suggestion in agreement with our other reviewer, we will replace with "resurgence point".

Note that e.g. in P5L91, mu+/- are being referred to as "collapse points".

R: We will replace "collapse points" with "bifurcation points".

P4L64: Can the authors elaborate why they think a double well potential has mainly been studied qualitatively? I would argue that this simple and general mathematical model has been studied quantitatively to an exceptional degree.

R: The reviewer is correct that this model has been extensively studied and applied. But to our knowledge it has not been quantitatively applied to AMOC collapse in complex numerical models or observational data in the way as we present. We will replace "studied mainly in a qualitative way (within catastrophe theory)" with "studied mainly qualitatively in connection with the Langevin equation." Bolton &a (Boulton, C., Allison,

[Figure]

L. & Lenton, T. Early warning signals of Atlantic Meridional Overturning Circulation collapse in a fully coupled climate model. Nat Commun 5, 5752 (2014).) do study an AMOC hysteresis loop qualitatively, but do not mention the Langevin equation. We will add Bolton &a to our references and discuss its relevance as a quantitative study of AMOC bistability. Specifically, the transient run studied in that paper and how it could relate to the Langevin model.

P4L65ff: It is a bit confusing when the authors first say that 2 parameters are enough to describe bi-stability, but then use another 2 parameters to scale and shift to the AMOC variable. Maybe it would be better to first explicitly say that by a shift and scale of the variable x, one can eliminate the third order term as well as the fourth order coefficient.

R: Correct, that is their purpose: to reduce the polynomial to a smaller set such that only the minimal number of parameters remain. We will add a clarification as suggested.

Both of these transformation do not influence the global bifurcation behavior. Then, they can state that a shift and scaling is considered when fitting to the climate model data.

R: True, because the topology is not affected. We will add a remark as suggested.

P4L78-81: Can the authors elaborate why they obtain these rough estimates for the parameters, and how they are insensitive to other parameter values?

R: These are not estimates but interpretations that can be linked to the bifurcation diagram. We will replace "The value of nu is roughly . . . " with "In the bifurcation diagram the value of nu is roughly . . . ." And likewise for lambda.

P5L90-91: When speaking about "solution" what exactly do the authors mean?

R: In this instance we mean that only the trivial solution exists: only 0 as the value for all variables.

P5L92-97: This section is a bit unclear. Can the authors define a "track", and what

does it mean to be one-dimensional?

R: We will remove the notion of a "track". It being 1-dimensional means that because the hysteresis loop is 1-dimensional the values (alpha, beta) as a function of mu are as well.

The fact that alpha and beta are called normal and splitting factor is better mentioned earlier.

R: Second mention to be removed. A more clear distinction of "parameter" and "variable" would be appropriate.

R: We will alter the text such that the Rahmstorf set has data mu and psi, alpha and beta are the stability parameters (which in turn are expressed as a rate and offset), nu and lambda are the scaling parameters, and mu+/- the bifurcation points.

P5L101: This argument is unclear to me. The fact that the AMOC is scalar variable should not constrain the path through the stability landscape in any way. Do the authors rather want to say that in the climate model experiments there is only a single control parameter mu, and that by assuming a linear dependency of both alpha and beta on mu, they can express some parameters by the extremal values of mu?

R: Indeed, mu is the only control parameter. By assuming linearity a reduced set of equations can be determined later on.

P8L127-129: Maybe the authors can elaborate more specifically on why these arguments are relevant in order to neglect a non-linear change of either mu or alpha/beta?

R: We will remove these lines, they are not needed for the argument.

P8L141-146: Improve this explanation. When introducing stochasticity, the asymptotic dynamics for each parameter value give rise to a stationary density.

R: Correct, to be added.

In the case of the scalar potential, this distribution can be given analytically up to a normalization factor. Thus, the distribution can be used as a likelihood function (if I understand correctly) for parameter inference with MCMC.

R: Correct, to be added.

P9L157: What is the "sampled" bi-modality region?

R: Sampled as in where the dataset has values. We will drop "sampled".

P9L160: "These changes correspond directly to the potential functions in Fig. 2." What is meant by this?

R: We mean that the distribution functions can be linked to the different characteristic shapes of the polynomial catalogued in fig 2. We will rewrite as "Each distinct shape of the distribution can be linked to one of the potential functions in Fig 2.".

P10L174: Can the authors explain why lambda < nu in general?

R: The offset (lambda) cannot exceed the scaling factor (nu) because the offset needs to be roughly in the middle of the two stable branches.

Caption Figure 5: The terminology is unclear. What is meant by a "singular" maximum?

R: We mean unimodality, to be rewritten as such.

What is meant by the dominant and the weak mode, and what is the inversion?

R: For a bimodal distribution there is a mode with more probability mass than the other which we call the dominant mode and the model with less mass the weak mode. And inversion is where these modes switch in strength: the dominant turn weak and the weak turns dominant.

There are also grammatical errors ("...in the middle and inversion from weak. . .").

R: "in the middle and inversion from weak to dominant takes place" to be removed from caption.

P11L180: I think a more precise statement would be that they estimate the posterior probability distribution of the parameters, given the data psi(mu). Furthermore, the following equation does not define the likelihood but the posterior distribution.

R: Yes, correct; will be rewritten as suggested: the equations states that the posterior distribution is proportional to the likelihood multiplies with the prior distribution.

P11L183: Why linearised?

R: To be removed, this related to linearisation of beta0, delta beta, etc

P11L184-188: This statement of Bayes' rule is not correct, please revise. The right hand side is not called Bayes' factor (which arises in model comparison).

R: We will remove L187-188.

P12L209-219: The constants muS+ etc. are not properly introduced and should be shown in one of the figures.

R: These are given in the caption of fig 6. We will add a reference to the figure on the text.

The footnote 2 needs to be explained better.

R: This is a technicality in how we defined the optimiser. We will rewrite to explain that this is useful to avoid solutions that intersect the B1 or B2 twice (see also P17L308-310 below).

Caption Table 1: Why is a linear function used and not a higher order polynomial? This does not seem to be very suitable to the data.

R: We only fitted the beginning (left part) of the upper branch and assumed a constant sigma. We will add this as a clarification to the text.

P13L228-230: This is not very precise wording. What do the authors mean with "unstable" and "more stable" solutions?

R: We will remove "more". Stability relates solely to the attractors and repellor.

P14L242-245: What do the authors want to say here? It comes as a surprise to me that suddenly only the data for mu > mu+ should be relevant?

R: We ignore the data on the lower branch before the collapse point because we did not want it to influence the fits, especially because we are only interested in the collapse from the upper branch.

And why do they now claim that the model C-GOLDSTEIN does not "appear" to show a collapse?

R: To be removed: our intent was to comment of the smoothness of the trajectory, but it is unnecessary,

P14L252-253: Unclear what the authors are trying to say.

R: We will remove these lines, they are redundant.

P14L259: What is meant by "non-linear degradation"?

R: We mean the part of the hysteresis loop after the collapse point: before that point the change was fairly linear, but after it is strongly non-linear.

P15L267: In what way is the model sufficient to describe the data? Certainly the "re-invigouration" is not well captured.

R: The aim was to model the collapse; the resurgence appears more difficult. We will add a clarification to emphasise this point and estimate a goodness of fit.

P17L308-310: Unclear what is meant here. What are "non-admissible" solutions?

R: An inadmissible solution is one where the curve through (alpha, beta) space intersects one of the subspace B1,2 twice. Because B1,2 are concave this is a possibility. To be discussed in the revised text.

P17L313-315: Unclear. Smoothing might be due to other reasons?

R: Perhaps, but is not mentioned in Rahmstorf (2005).

P17L323: What is meant by "direct numerical stochastic integration"?

R: This remark was originally intended to point out another way to perform the calculations: by solving the SDE directly. We now believe this remark to be redundant and will remove it.

P17L330: How exactly does this paper present a step forward to assess the likelihood of a future collapse of the AMOC? The method presented here relies on previously modeled collapses of the AMOC with realistic climate models. How does the method generate additional information?

R: If the characterisation has predictive values, more complex models can be used to derive a collapse point if the freshwater forcing at the bifurcation points can be estimated. It is also a way to compare the collapse characteristics of various models. If the freshwater forcing at the bifurcation points can be generalised and linked to a robust indicator (such as, perhaps, Mov), the method can be applied to the real world as well. We agree, there are quit some steps in between the method outlined here and its extension to the modelled and observed timeseries. We will expand the discussion on this point.

Technical corrections

R: We are in agreement with all corrections below

P1L15: last glacial maximum and early holocene -> last glacial period

R: To be replaced as suggested.

P2L26: . . . which are presumed to be functions . . .

R: To be corrected.

C6P2L30: ...and tipping points in the climate, it has not been . . .

R: To be corrected.

P2L38: . . . or an increased surface freshwater flux by changes in precipitation minus evaporation. . .

R: To be corrected.

P2L46: scalar variable obtained by integrating . . . To be corrected.

P2L52: . . . one of the two basins of attraction vanishes . . .

R: To be corrected.

P7L126: rather "(delta alpha, delta beta)"?

R: Correct: changes in alpha, beta: to be corrected. P8L142: Remove: As shown by Cobb (1978), this distribution belongs to the exponen- tial family.

R: We will remove this.

P8L144: The polynomial potential introduced in the previous section, we had already. . ., gives the probability . . .

R: To be corrected.

P8L148: Note that C = C(alpha,beta), which does not have. . .

R: To be corrected.

P8L150: . . . because of the scaling . . .

R: To be corrected.

P8L152 and Fig. 4 caption: In what way is this a sample collapse trajectory, or an example trajectory?

R: This should indeed be example, not sample.

P9L161: . . . a change in only one . . .

R: To be corrected

P9L163-165: Why not say this at P8L148? It is a bit redundant otherwise.

R: We will move these lines to P8L148.

P10L168: arrived at -> described

R: To be corrected

P10L171: independent of each other

R: To be corrected.

P11L180: The method used is not considered machine learning.

R: To be removed.

P11L190: Does not seem to be relevant, as it is not done here.

R: This sentence relates to "These resultant posterior distributions can, in turn, be used as prior distributions, yielding a chain of sampled parameter vectors." It is roughly how the sampler works, but we will remove this because it is redundant.

P11L194-196: This is partly redundant, and it is not clear why the authors mean that the model can be fit with uninformative priors.

R: We will remove the part about uninformative priors: it is unnecessary.

P11L199-200: Redundant.

R: We will replace "With $\nu$ and $\lambda$ introduced earlier, the state variable x undergoes an affine transformation and normalises the polynomial. These ..." with "The parameters $\nu$ and $\lambda$ ... "

P11L207: Redundant.

R: To be removed.

P11L207-208: An overview of priors is: The following prior distributions are used:

R: To be corrected.

P14L243: do -> to

R: To be corrected.

P14L254: Why "sample" paths? The authors are showing distributions, which is exactly contrary to showing sample paths.

R: We will remove " . . . for linearly parametrised sample paths through the stability space" . . . "

P14L255: Redundant.

R: We will remove "The $\beta$ parameter changes linearly and $\alpha$ follows from the constraints in Eqs. 4 and 5. Blue and red lines indicate the prior bounds for $\mu -$ and $\mu +$, respectively."

P15L265: couples -> models. Why "additionally"?

R: This is not needed: we will remove "Additionally, a linear parameterisation through state space couples to the freshwater applied to the North Atlantic subtropical gyre region."

P16L275: Do they mean arg(max(|psi|)) ?

R: We mean max(|psi|), to be corrected.

P17L309 till its -> until it

R: To be corrected.
* * *

---

## Author Response (AR1)

[revised manuscript text omitted]

[3]The figure we used is a vector graphic and the data set can be retrieved from it by inverting the plot matrices used to map the original data to the values in the graph. We can replicate the data in this manner.

[Figure]

**Figure 6.** Absolute values of numerical derivatives (left) from the trajectories of AMOC strength as function of freshwater forcing to the right (taken from Rahmstorf et al. (2005, Fig. 2, bottom panel), reproduced with permission from the publisher: American Geophysical Union). In red the upper branch, blue the lower branch. Left column: Bremen, ECBilt-CLIO, C-GOLDSTEIN; right column: MOM hor, MOM iso, UVic. Vertical solid lines mark $\mu = 0$ (blue) and $\mu = 0.2$ (red); vertical dashed lines mark the chosen boundary values for $\mu_{\pm}$. All values have units Sv.

| model | #data points | ocean component | atmosphere component | reference |
|---|---|---|---|---|
| Bremen | 2461 | large-scale geostrophic | energy balance | Prange et al. (2003) |
| ECBilt-CLIO | 243 | 3D primitive equations | quasi-geostrophic | Goosse et al. (2001) |
| C-GOLDSTEIN | 849 | 3D simplified | energy-moisture balance | Edwards and Marsh (2005) |
| MOM hor | 1233 | 3D primitive equations (MOM) | simple energy balance | Rahmstorf and Willebrand (1995) |
| MOM iso | 1442 | as above, with isopycnal mixing | simple energy balance | |
| UVic | 464 | 3D primitive equations (MOM) | energy-moisture balance | Weaver et al. (2001) |

**Table 1.** Overview of models used. Each data point is independent from the others because each is the result of a quasi steady state run. The number of data points used is given. The summary of the type of model component and references are taken from Rahmstorf et al. (2005).

| model | $\sigma$ | $\mu_-$ | $\mu_+$ | present da|
|---|---|---|---|---|
|  Bremen |  0.181 | [ -0.018, 0.010] | [ 0.120, 0.220] | (  0.070( |
|  ECBilt-CLIO |  0.176 | [ -0.044, 0.030] | [ 0.115, 0.210] | (0.050, 22.8)  -0.010, 0.010 0.1|
| C-GOLDSTEIN | 0.122 | [-0.100, 0.035] | [0.115, 0.190] | (-0.100, 29.0|
|  MOM hor |  0.526 | [ 0.010] | [ 0.130, 0.200] | (  0.11|
|  MOM iso |  0.216 | [ -0.010, 0.020] | [ 0.210] | (  +0.0|
| UVic | 0.260 | [-0.020, 0.010] | [0.188, 0.225] | ( 0.080, 25.0|

**Table 2.** Overview of models, the estimated standard deviation with the upper branch fitted to a linear function (note that the original trajectories had already been smoothed), the ranges of $\mu_\pm$, the location of present day in the models, and whether the present day value is in the unimodal regime (+) or not (-). All values  have units  Sv.

If no other mechanisms apart from the salt advection are important we expect the bifurcation points to lie beyond the observed transition points because a noise-induced transition pushes the AMOC into the off-state sooner. (Note that although the collapse points are expected to lie before these peaks, low levels of noise will obscure this effect.) The dashed lines indicate the regions where we will search for the optimum values of $\mu_\pm$. These differ from the fixed 0 and 0.2 values chosen by (Rahmstorf et al., 2005), who also shifted the trajectories to align on these values.

[revised manuscript text omitted]

**4 Discussion and conclusion**

We derived a simple model of AMOC collapse based on Langevin dynamics (Eq. 1) with a changing freshwater forcing ($\mu$) and applied this to EMIC simulated collapse trajectories taken from Rahmstorf et al. (2005). The collapse occurs at a bifurcation point $\mu_+$ which appears smaller than given in (Rahmstorf et al., 2005). A corresponding bifurcation point $\mu_-$ relates an abrupt transition back to the on-state.  The AMOC also requires an offset and scaling parameter to be fitted ($\lambda$ and $\nu$). These six parameters are sufficient to describe the abrupt collapse  of the AMOC that leads to a hysteresis loop under varying freshwater forcing. The resurgence of the AMOC is not the same as the collapse process and we did not attempt to obtain an accurate fit of that part of the hysteresis loop.

[revised manuscript text omitted]
" is an interesting application of the classic analytical approach of Poston and Stewart with introduced stochasticity for modelling AMOC trajectories of the EMICs published in [Rahmstorf et al 2005]. I think the paper should be published after a minor revision.

R: We thank the reviewer for helpful comments and suggestions. Below you can find our responses.

The title should be corrected: "Modelling collapse of the Atlantic Meridional Overturning using the Langevin dynamics".

[x]R: Our suggestion would be "Characterisation of Atlantic Meridional Overturning hysteresis using Langevin dynamics" to emphasise the purpose of the paper better, that is, using a reduced set of numbers to quantitatively describe the AMOC collapse under a freshwater forcing.

As the authors admit themselves, EMICs are not sufficiently, representative of the real climate. Also, given the number of parameters the authors use to fit their model (six) and their geometrical origin (see description of ν and λ), I understand why the authors claim that only the freshwater forcing is the variable that determines the dynamical b

[x]R: Unfortunately, here seems to be a typesetting problem at ESD that renders some of the comments to be unreadable. As we understand it, the question is about using only freshwater as forcing for studying AMOC stability. The other possible forcing effect is thermal, and in principle a sufficiently large warming could also halt deep water formation and induce a collapse of the AMOC. However, in this paper we intend to explain the hysteresis behaviour shown in Rahmstorf et al. (2005), which is obtained by changing the freshwater forcing. As a result, we use this forcing as the dynamical variable that controls the stability regime of the AMOC. This point is discussed in the text now.

It would be interesting to see how the model can be used for forecast of bifurcations. The authors perform derivation of the model parameters using Bayesian framework, but once the model has been fully formed and the parameters are obtained for several EMICs, can the authors attempt forecast or hindcast of the bifurcating time series?

[x]R: We thank the reviewer for this interesting comment which could be explored in further research. A forecast from a partial AMOC weakening series would require an estimate of future freshwater forcing, and maybe making use of EMIC (or GCM) derived values as estimates. We added a paragraph in the Discussion section where we consider options along these lines for future research.

[Rahmstorf et al 2005] paper used 11 models and only hysteresis loops were presented (not actual AMOC trajectories)
https://agupubs.onlinelibrary.wiley.com/doi/pdfdirect/10.1029/2005GL023655?download=true

[x]R: Our calculations are very time consuming. For this reason we decided to focus on the 6 models with most complete representations of the physics amongst the numerical models and disregard the 5 models without a 3-D ocean component. We consider that their characterisation is too far removed from the real world or CMIP class numerical models. This is now explained in the text (L227).

Can a figure be added with plotted time series that could be derived from the obtained model? For example, for the set of parameters averaged over a set of the selected EMICs? I wonder how realistic could be the time series and at what time scale it could forecast an AMOC bifurcation?

[x]R: This is a good suggestion, but unfortunately no timeseries were given in the published data. To derive those new runs would have to be made. The hysteresis loops are obtained by changing the forcing with small steps and then obtaining a new (quasi) equilibrium state for the changed forcing.

I understand that the framework is quite heavy computationally. Can the authors add discussion on how applicable can be this approach in other areas of geosciences where similar potential models may be used?

[x]:R: In principle, any hysteresis curve that is produced under a forcing where the lambda and nu transformations suffice to normalise the curve could be used. The calculation is indeed quite heavy in computational terms, but not more time-consuming for a hysteresis curve obtained in a full Earth System Model than for a hystresis curve from a much simpler EMIC. Other geophysical processes might be icesheet mass loss (e.g. Robinson, Alexander & Calov, Reinhard & Ganopolski, Andrey. (2012). Multistability and critical thresholds of the Greenland Ice Sheet. Nature Climate Change. 2.), forest dieback (e.g. Staal, A., Dekker, S.C., Xu, C. *et al.* Bistability, Spatial Interaction, and the Distribution of Tropical Forests and Savannas. *Ecosystems* **19,** 1080–1091 (2016)), or lake turbidity (Scheffer, M., van Nes, E.H. Shallow lakes theory revisited: various alternative regimes driven by climate, nutrients, depth and lake size. *Hydrobiologia* **584,** 455–466 (2007). Any process which allows two stable states with rapid transitions between them and an asymmetric response to the forcing could be described by our method.

Paragraph added (2$^{nd}$) in discussion.

The authors derived datasets from the published figures - is it allowed practice? Shouldn't they be obtained from the authors as datasets? Can the authors add information about the derived datasets in the table (number of points, etc)? Also, can more recent EMICs be used?

[x]R: We strongly support the development of open science and making data findable and accessible. Unfortunately, we have not been able to obtain the original datasets from the authors (we received no reply to our requests). The (individual points of the) measured values were retrieved from the plots by inverting the transformation matrices. This can be done for certain plots that are converted to pdf from plotting software such as Matlab. The dataset is then numerically the same as the set used to produce the plots in Rahmstorf et al (2005). The publisher allows for the use of individual graphics from their publications: we will note this in the text (L233).

[x]We have included a table with additional information about the dataset.

[x]In principle any hysteresis curve can be used, but we have not expanded the data set beyond the Rahmstorf et al set.

Further comments

The abstract should be modified to say that model is fitted to the trajectories.

[x] R: Corrected.

In the first paragraph, AMOC acronym is introduced twice.

[x] R: 2nd mention removed.

Instead of "invigoration" it is better to say "re-activation".

[x] R: Corrected with "resurgence" as suggested by our other reviewer.

Line 90 – "diagrams"

[x] R: Corrected.

Figure 2 - labels in all panels should be of the same font size

[x]R: Replotted with the same label size.

Line 124 - "the simplest"

[x] R: Corrected.

Line 152 - grey lines are mentioned in Figure 4, not clear which, maybe make them dashed? Similarly, dashed lines in Figs. 6,7 are impossible to see - enlarge these figures and all labels.

[x] R: Replotted with a colour different from grey and enlarge the labels.

Table 1 should be expanded to include more information on the selected models - countries, resolution, etc.

[x] R: Table included.

This is an interesting paper, which aims to describe the collapse and hysteresis of the AMOC observed in intermediate complexity climate models subject to freshwater forcing by low-dimensional Langevin dynamics as a stochastic bifurcation of a double well potential. Substantial revisions are necessary to improve the clarity of the manuscript and to support the conclusions.

R: We thank the reviewer for the detailed comments and valuable suggestions. Below are our revised comments and performed alternations.

General comments
1.) It is not clear what the purpose of the paper is. The authors do not state what their model is able to explain or predict.
Is the purpose to predict the exact parameter value of a collapse? Or at least to develop a method to do this?

[x] R: Our aim is to investigate whether it is possible to model the outcome of complex numerical models with a low dimensional model and thereby enhance understanding of the physics of an AMOC collapse and its hysteresis behaviour. The Langevin model defines a low-dimensional manifold that captures the essential collapse characteristics. To the extent that the low-dimensional model is successful in capturing the more complex model this investigation can indeed be seen as the development of a method to predict the parameter range where in a model a collapse would occur. The method is thus partly geared toward providing a means of prediction, but (at present) mainly to provide some characterisation of the collapse that will allow comparison between climate models. If a good fit can be found, then we can further explain the non-dynamical nature of the AMOC variations. As we show, this is partially the case. We have stated in the introduction where the purpose of the paper is defined.

Are there prospects to apply the method to observational data?

[x] R: Since the forcing is a freshwater anomaly in the North Atlantic, we would need to estimate the counterpart in the real world. Moreover, the forcing values at the bifurcations have to be known. At present, this is not the case for the real world, and in models it is model-dependent. An attempt could be made to relate the forcing to an indicator that can be linked to the bifurcation points, for instance $M_{ov}$, the freshwater transport by the overturning. This should first be tested in more comprehensive numerical models before applying it to observations. If this can be done, then from the transient change in mu and with knowledge of sigma from observed AMOC variations, a predictive model for the likelihood of an AMOC collapse could be developed. We have added a paragraph in the Discussion section with an outlook of future research.

Or is an aim to understand dynamically what is happening in realistic climate models?
This should be stated in the introduction

[x] R: We would like to see this paper as a first step to say more about the behaviour of the AMOC in complex numerical climate models such as used in the CMIP model intercomparisons. In particular, we propose a simplified low-dimensional model that is able to explain (and predict) bifurcation (tipping) points and abrupt change in the AMOC. Ultimately, this could be used to investigate abrupt behaviour in CMIP models, and how likely abrupt changes would be, even if not simulated by the model. Without a-priori knowledge of the freshwater-forcing values associated with the model's bifurcation points, first must be investigated whether mu can be linked to a general indicator like Mov (see the comment above). We have stated the purpose of the paper more clearly in the introduction and add the outlook above to the discussion section.

(P2L29ff). It is also unclear whether they want to only/mostly model the AMOC collapse (as stated at some points in the paper) or also the resurgence.

[x] R: We are mainly interested in modelling the collapse, not the resurgence. We note this in the manuscript as "Although the hysteresis loops of the AMOC include both a collapse and a resurgence, we will only attempt to model the collapse from the stable on-branch to the stable off-branch."

2.) Regarding the conclusions, how can the authors say that the model successfully captures the dynamics?
They don't compare with other models of higher or lower complexity, nor do they have any metric that shows goodness of fit or anything similar. This would be necessary to make such a conclusion.

[x] R: We thank the reviewer for this comment. We will add the posterior spread to show the goodness of fit. To explore and compare with other low-dimensional models than the Langevin model is beyond the scope of this paper (the six included already form a multi-model ensemble). A possible next step could be to apply the Langevin model to a transient run where mu depends on time. The AMOC should eventually collapse when increasing the freshwater forcing and we can have greater confidence in the applicability of our approach after testing it to such a transient simulation.

3.) The manuscript is not very well written and hard to follow. The terminology is often unclear. (E.g. what is a "track", and how does the use of "stability landscape" apply here? See specific comments.) Some corrections are given under "technical corrections", but the language and terminology has to be generally improved throughout the manuscript. Furthermore, I believe the manuscript can be shortened severely. What the authors want to get across can be said more efficiently. Many things are mentioned twice or more (see specific and technical comments for suggestions). Finally, the labels in multiple figures are unreadable.

[x] R: We critically reviewed the text. We comment on identified issues with the specific comments below. In short, the terminology has been clarified, and parts of the text that were confusing have been removed. The figures have been improved with larger and consistent fonts for the labels.

4.) The data acquisition seems problematic. I am not sure whether it is viable for this journal to present a data analysis based on visually extracted data from a figure of another publication.
Accordingly, the quality of the data is a major drawback of the study (e.g. arbitrary smoothing and AMOC metric).
Their main problem in fitting the data might be due to the specific metric that is shown in the Rahmstorf et al. (2005) figures, so it is a shame that the authors are not able to resolve that.

[x] R: Indeed, the data has been obtained from the figures of the paper. However, it is not visually extracted as the reviewer suggests. The figure we used is a vector graphic and the dataset can be retrieved from it by inverting the plot matrices used to map the original data to the values in the graph. We can exactly replicate the data used for this figure in this manner. In order to validate this method, we asked for the original data from Rahmstorf et al. (2005) but have not received a reply at present.
Less smoothing, and presumably larger noise levels, would likely show a stochastic collapse more easily.
We fully agree with this remark but want to emphasize that the main goal here is to develop and test a method to capture complex model behaviour with a simple low-order model. The methodology

described here is not affected by smoothing, only the assessment how well the method works is somewhat hindered by this.

5.) The description of their method contains many errors, and is incomplete. An explicit expression for the likelihood, as well as details of the Metropolis-Hastings implementation are missing.

[x] R: We agree that more detail could be given, but we believe a description of the Metropolis-Hastings algorithm is too much detail for this paper as it is well established and already described in many textbooks. We added a reference to the textbook of Bernardo & Smith which describes this algorithm.

In the discussion, the authors name difficulties in the numerical implementation as a possible reason for the failure of their fit to describe the lower AMOC branch, but it is for the reader not possible to assess whether this is relevant, since no details or robustness tests are given.
Furthermore, it is not stated how many data points the respective data sets contain, and it is not mentioned that the authors assume successive data points to be independent.

[x]R: Indeed, it is important to assess the robustness of our implementation. To further detail the validity methodology and outcome, we added a table with model characteristics and state that each point is independent.
Bremen: 2461
EC-Bilt-CLIO: 243
C-GOLDSTEIN: 849
MOM hor: 1233
MOM iso: 1442
UVIC: 464
In addition, we want to emphasize that our main goal is to model the transition from on-branch to off-branch, that is, the upper right half of the hysteresis curve, and not so much the dynamics that govern the lower branch, also because we assume that other dynamics govern the lower branch and our simple model has to be extended to account for those dynamics.

It is also not mentioned how the maximum of the posterior parameter distributions is picked.

[x]R: The most likely value is the mean of the posterior distribution. This does assume that the posterior distributions are unimodal. We will discuss this in the text.

6.) Finally, several questions regarding the methodology.
 a) Why do the authors not try to estimate sigma with their Bayesian method?

[x]R: In principle this can be done. The variation in the hysteresis loops appears constant and can therefore be estimated more easily by other means. This does add to the computational costs and expands the search space, however, making it more difficult to find solutions. Therefore, we did not follow this method. We will mention this in the text (L261).
Why not include observational noise?

[x]R: Observational noise of the real AMOC would have a larger spread than the sigma we obtained. Synthetic series on the basis of the found parameters could indeed be generated with such a noise level. However, we intend to fit the intermediate complexity model outcome, using the data of that particular model. AMOC collapse and its likelihood at a given point in parameter space is model-dependent. One of the essential parameters in this dependency, is the model-dependent

sigma. Therefore, we prefer to use the sigma that is characteristic of each particular model. This point is now discussed in the paper (L257).

This could handle the fact that the data is filtered arbitrarily. It could also completely change the locations of the inferred bifurcation points.

[x] R: The bifurcation points are determined by the limit (non-stochastic) solutions. A noise driven transition could occur, however, and push the points that bound the hysteresis curve further inwards, towards each other. We will mention this limitation more clearly in the paper (L241).

 b) To make the paper more understandable it would be good to note explicitly early in the manuscript that the movement of mu is actually known.

[x]R: We agree with the reviewer. We added "The forcing values of mu are known and the same for each climate model."
However, the values of mu+ and mu- are not, and model-dependent.

c) Why not try multiplicative noise? (see also e.g. Das/Kantz Phys. Rev. E 101, 062145, 2020) This should relatively easily give a model that describes the asymmetric behavior.

[x] R: Multiplicative noise is state dependent, while we have made the assumption that the noise level is constant; therefore, we used additive noise. We will discuss this point in the text (L160).

d) It should be noted explicitly that there is no time dependency of the data. I wonder why they choose not to fit to time series instead? This would allow to treat the non-equilibrium nature of the data. Also, it would be much more applicable to observational data and to make predictions.

[x]R: Indeed, we will add a remark that each data point corresponds to a fixed value of mu and is not time dependent. No timeseries are available, however. Each mu value represents a separate climate model run which has run (more or less) to equilibrium. We need to be able to estimate mu+ and mu- from the models, and this can only be done from a hysteresis-curve which indeed contains equilibrium solutions and no time-dependence. We agree that a logical next step is to apply the method on time-dependent runs, but to validate the model it is needed that we know the equilibrium bifurcations points as well for that model, and that the time-dependent runs are based on a slowly-changing mu and include an AMOC collapse. There are not many models available that answer all these requirements. We added a paragraph in the Discussion section where we mention this point.

e) Why not only move along alpha at a certain fixed beta? Is moving both parameters supported by the data significantly better?

[x]R: Early attempts with a fixed beta resulted in worse fits; therefore, we opted not to restrict beta. Removed (see comment P14L255)

Specific comments
Abstract: "Machine learning": To my knowledge MCMC is not considered a machine learning technique. The abstract needs to be expanded to better reflect the motivation of the study, what their method enables them to do, and their conclusions.

[x] R: We will remove mentioning machine learning. We will also add to the abstract: "The Langevin model allows for comparison between models that display an AMOC collapse. Variation between climate models studied here is mainly in the strength of the stable AMOC and the strength of the response to a freshwater forcing."

P2L42-45: This is a not a very clear explanation of the salt advection feedback. The main point is that North Atlantic salinity anomalies (positive/negative) are amplified by their effect on the overturning flow (strengthening/weakening), the strength of which controls the North Atlantic salinity.
This is thus a positive feedback and leads to bi-stability with the associated possibility of abrupt transitions.

[x]R: Rewritten as suggested.

P2L53: ". . .number of solutions for a given value of the freshwater forcing goes from 2 to 3. . . ". Should say "goes from 3 to 1" as the bifurcation point is crossed. (There are 2 solutions precisely at the bifurcation point, but I think this saddle-node fixed point is not relevant here.)

[x] R: Corrected this as suggested.

P3 Caption Fig.1: The terminology of this figure is not appropriate and furthermore not understandable at this point within the manuscript. No trajectory is shown, but a bifurcation diagram.

[x] R: Correct,  renamed to "bifurcation diagram".

They have to be more specific with what they mean by a deformation of the "trajectory".
Also, at this position within the manuscript, it is completely unclear what they mean with "trench of the distribution".
Either leave out or explain in the main text.

[x] R: Use of "trench" removed from the text.

Furthermore, I suggest to use the term "resurgence point" for mu-, and use that terminology throughout the paper.

[x] R: Suggestion in agreement with our other reviewer, replaced with "resurgence point".

Note that e.g. in P5L91, mu+/- are being referred to as "collapse points".

[x] R: Replaced "collapse points" with "bifurcation points".

P4L64: Can the authors elaborate why they think a double well potential has mainly been studied qualitatively? I would argue that this simple and general mathematical model has been studied quantitatively to an exceptional degree.

[x] R: The reviewer is correct that this model has been extensively studied and applied. But to our knowledge it has not been quantitatively applied to AMOC collapse in complex numerical climate models or observational data in the way as we present here. We will replace "studied mainly in a qualitative way (within catastrophe theory)" with "the double well potential has been extensively studied and applied, also in a quantitative way. But to our knowledge it has not been quantitatively applied to AMOC hysteresis using the Langevin equation in complex numerical climate models before.
Bolton et al (Boulton, C., Allison, L. & Lenton, T. Early warning signals of Atlantic Meridional Overturning Circulation collapse in a fully coupled climate model. *Nat Commun* **5,** 5752 (2014).)

do study an AMOC hysteresis loop qualitatively, but do not mention the Langevin equation. We added Bolton et al to our references and discuss its relevance as a quantitative study of AMOC bistability, specifically, the transient run studied in that paper and how it could relate to the Langevin model.

P4L65ff: It is a bit confusing when the authors first say that 2 parameters are enough to describe bi-stability, but then use another 2 parameters to scale and shift to the AMOC variable.
Maybe it would be better to first explicitly say that by a shift and scale of the variable x, one can eliminate the third order term as well as the fourth order coefficient.

[x] R: The reviewer is correct, indeed that is their purpose: to reduce the polynomial to a smaller set such that only the minimal number of parameters remain. Added clarification as suggested (L77).

Both of these transformation do not influence the global bifurcation behavior. Then, they can state that a shift and scaling is considered when fitting to the climate model data.

[x] R: The reviewer is right, because the topology is not affected. We added a remark added as suggested (L92).

P4L78-81: Can the authors elaborate why they obtain these rough estimates for the parameters, and how they are insensitive to other parameter values?

[x] R: These are not estimates but interpretations that can be linked to the bifurcation diagram. We will replace "The value of nu is roughly … " with "In the bifurcation diagram the value of nu is roughly … ." And likewise for lambda.

P5L90-91: When speaking about "solution" what exactly do the authors mean?

[x] R: In this case we mean that only the trivial solution exists: only 0 as the value for all variables. This is further explained in the text (L104).

P5L92-97: This section is a bit unclear. Can the authors define a "track", and what does it mean to be one-dimensional?

[x] R: We removed the notion of a "track". It being 1-dimensional means that because the hysteresis loop is 1-dimensional the values (alpha, beta) as a function of mu are as well.

The fact that alpha and beta are called normal and splitting factor is better mentioned earlier.

[x] R: Sentence moved up.

A more clear distinction of "parameter" and "variable" would be appropriate.

[x] R: We have clarified the text such that the Rahmstorf set has data mu and psi; alpha and beta are the stability parameters (which in turn are expressed as a rate and offset); nu and lambda are the scaling parameters, and mu+/- the bifurcation points.

P5L101: This argument is unclear to me. The fact that the AMOC is scalar variable should not constrain the path through the stability landscape in any way.

Do the authors rather want to say that in the climate model experiments there is only a single control parameter mu, and that by assuming a linear dependency of both alpha and beta on mu, they can express some parameters by the extremal values of mu?

[x] R: Indeed, mu is the only control parameter. By assuming linearity a reduced set of equations can be determined later on.  This is now explained in the text (L139).

P8L127-129: Maybe the authors can elaborate more specifically on why these arguments are relevant in order to neglect a non-linear change of either mu or alpha/beta?

[x] R: We will remove these lines, they are not needed for the argument.

P8L141-146: Improve this explanation. When introducing stochasticity, the asymptotic dynamics for each parameter value give rise to a stationary density.

[x] R: The reviewer is correct, we added "The potential function can be replaced by a distribution which is the stationary distribution in the asymptotic limit (i.e. the long term behaviour of repeated sampling of the hysteresis loop).".

In the case of the scalar potential, this distribution can be given analytically up to a normalization factor. Thus, the distribution can be used as a likelihood function (if I understand correctly) for parameter inference with MCMC.

[x] R: The reviewer is correct, we added. " but can be computed numerically (and therefore used as a likelihood function in the next section)"

P9L157: What is the "sampled" bi-modality region?

[x] R: Sampled as in where the dataset has values. We  dropped "sampled".

P9L160: "These changes correspond directly to the potential functions in Fig. 2." What is meant by this?

[x] R: We mean that the distribution functions can be linked to the different characteristic shapes of the polynomial catalogued in fig 2. We will rewrite as "Each distinct shape of  the distribution can be linked to one of the potential functions in Fig 2.".

P10L174: Can the authors explain why lambda < nu in general?

[x] R: The offset (lambda) cannot exceed the scaling factor (nu) because the offset needs to be roughly in the middle of the two stable branches.

Caption Figure 5: The terminology is unclear. What is meant by a "singular" maximum?

[x] R: Removed (superfluous)

What is meant by the dominant and the weak mode, and what is the inversion?

[x] R: For a bimodal distribution there is a mode with more probability mass than the other which we call the dominant mode and the model with less mass the weak mode. And inversion is where these modes switch in strength: the dominant turn weak and the weak turns dominant. Replaced with "small" and "large".

There are also grammatical errors ("...in the middle and inversion from weak. . .").

[x] R: "in the middle and inversion from weak to dominant takes place" removed from caption.

P11L180: I think a more precise statement would be that they estimate the posterior probability distribution of the parameters, given the data psi(mu).
Furthermore, the following equation does not define the likelihood but the posterior distribution.

[x]R: Yes, the reviewer is correct; rewritten as suggested: the equations states that the posterior distribution is proportional to the likelihood multiplies with the prior distribution.

P11L183: Why linearised?

[x]R: Removed, this related to linearisation of beta0, delta beta, etc

P11L184-188: This statement of Bayes' rule is not correct, please revise. The right hand side is not called Bayes' factor (which arises in model comparison).

[x]R: We removed L187-188.

P12L209-219: The constants muS+ etc. are not properly introduced and should be shown in one of the figures.

[x] R: These are given in the caption of fig 6. Reference to the figure given in the text.

The footnote 2 needs to be explained better.

[x] R: This is a technicality in how we defined the optimiser. Rewritten to explain that this is useful to avoid solutions that intersect the B1 or B2 twice (see also P17L308-310 below).

Caption Table 1: Why is a linear function used and not a higher order polynomial? This does not seem to be very suitable to the data.

[x] R: We only fitted the beginning (left part) of the upper branch and assumed a constant sigma. clarification to the text added (L255).

P13L228-230: This is not very precise wording. What do the authors mean with "un-stable" and "more stable" solutions?

[x] R: Removed "more". Stability relates solely to the attractors and repellor.

P14L242-245: What do the authors want to say here? It comes as a surprise to me that suddenly only the data for mu > mu+ should be relevant?

[x] R: We ignore the data on the lower branch before the collapse point because we did not want it to influence the fits, especially because we are only interested in the collapse from the upper branch. This is now mentioned in the text (L32,L249).

And why do they now claim that the model C-GOLDSTEIN does not "appear" to show a collapse?

[x] R: Removed: our intent was to comment of the smoothness of the trajectory, but it is unnecessary,

P14L252-253: Unclear what the authors are trying to say.

[x] R: We removed these lines, they are redundant.

P14L259: What is meant by "non-linear degradation"?

[x] R: We mean the part of the hysteresis loop after the collapse point: before that point the change was fairly linear, but after it is strongly non-linear. Removed

P15L267: In what way is the model sufficient to describe the data? Certainly the "re-invigouration" is not well captured.

[x] R: The aim was to model the collapse; the resurgence appears more difficult. Added a clarification to emphasise this point and estimate a goodness of fit.

P17L308-310: Unclear what is meant here. What are "non-admissible" solutions?

[x] R: An inadmissible solution is one where the curve through (alpha, beta) space intersects one of the subspace B1,2 twice. Because B1,2 are concave this is a possibility. Removed from text

P17L313-315: Unclear. Smoothing might be due to other reasons?

[x] R: Perhaps, but is not mentioned in Rahmstorf (2005).

P17L323: What is meant by "direct numerical stochastic integration"?

[x] R: This remark was originally intended to point out another way to perform the calculations: by solving the SDE directly. We now believe this remark to be redundant and removed it.

P17L330: How exactly does this paper present a step forward to assess the likelihood
of a future collapse of the AMOC?
The method presented here relies on previously modeled collapses of the AMOC with realistic climate models.
How does the method generate additional information?

[x] R: If the characterisation has predictive values, more complex climate models can be used to derive a collapse point if the freshwater forcing at the bifurcation points can be estimated. It is also a way to compare the collapse characteristics of various models. If the freshwater forcing at the bifurcation points can be generalised and linked to a robust indicator (such as, perhaps, Mov), the method can be applied to the real world as well. We agree, there are quit some steps in between the method outlined here and its extension to the modelled and observed timeseries. We will expand the discussion on this point.

Technical corrections

[x] R: We are in agreement with all corrections below

P1L15: last glacial maximum and early holocene -> last glacial period

[x] R: Replaced as suggested.

P2L26: . . . which are presumed to be functions . . .

[x] R: Corrected.

C6P2L30: ...and tipping points in the climate, it has not been . . .

[x] R: Corrected.

P2L38: . . . or an increased surface freshwater flux by changes in precipitation minus evaporation. . .

[x] R: Corrected.

P2L46: scalar variable obtained by integrating …
[x] Corrected.

P2L52: . . . one of the two basins of attraction vanishes . . .

[x] R: Corrected.

P7L126: rather "(delta alpha, delta beta)"?

[x] R: Correct: changes in alpha, beta: corrected.
P8L142: Remove: As shown by Cobb (1978), this distribution belongs to the exponential family.

[x] R: We removed this.

P8L144: The polynomial potential introduced in the previous section, we had already. . ., gives the probability . . .

[x]R: Corrected.

P8L148: Note that C = C(alpha,beta), which does not have…

[x]R: Corrected.

P8L150: . . . because of the scaling …

[x] R: corrected.

P8L152 and Fig. 4 caption: In what way is this a sample collapse trajectory, or an example trajectory?

[x] R: This should indeed be example, not sample.

P9L161: . . . a change in only one . . .

[x] R: Corrected

P9L163-165: Why not say this at P8L148? It is a bit redundant otherwise.

[x] R: We moved these lines to P8L148.

P10L168: arrived at -> described

[x] R: Corrected

P10L171: independent of each other

[x] R: Corrected.

P11L180: The method used is not considered machine learning.

[x] R: Removed.

P11L190: Does not seem to be relevant, as it is not done here.

[x] R: This sentence relates to "These resultant posterior distributions can, in turn, be used as prior distributions, yielding a chain of sampled parameter vectors." It is roughly how the sampler works, but we removed this because it is redundant.

P11L194-196: This is partly redundant, and it is not clear why the authors mean that the model can be fit with uninformative priors.

[x] R: We removed the part about uninformative priors: it is unnecessary.

P11L199-200: Redundant.

[x] R: We replaced "With $\nu$ and $\lambda$ introduced earlier, the state variable $x$ undergoes an affine transformation and normalises the polynomial. These ..." with "The parameters $\nu$ and $\lambda$ … "

P11L207: Redundant.

[x] R: Removed.

P11L207-208: An overview of priors is: The following prior distributions are used:

[x]R: Corrected.

P14L243: do -> to

[x]R: Corrected.

P14L254: Why "sample" paths? The authors are showing distributions, which is exactly contrary to showing sample paths.

[x]R: We removed " … for linearly parametrised sample paths through the stability space" … "

P14L255: Redundant.

[x]R: We removed "The β parameter changes linearly and α follows from the constraints in Eqs. 4 and 5. Blue and red lines indicate the prior bounds for μ − and μ + , respectively."

P15L265: couples -> models. Why "additionally"?

[x]R: This is not needed:  we removed "Additionally, a linear parameterisation through state space couples to the freshwater applied to the North Atlantic subtropical gyre region."

P16L275: Do they mean arg(max(|psi|)) ?

[x] R: We mean max(|psi|),  corrected.

P17L309 till its -> until it

[x] R: Sentence removed

---

## Referee Report (RR1)

Thank you for the opportunity to review this manuscript again. While it would have been nice for the authors to address several points more explicitly, which would have helped to increase the scope of the paper, the corrections helped to make the text more readable and make the purpose of the paper more clear.

I have a few additional comments:

**1.**
Regarding how they can substantiate that the model successfully captures the dynamics, the authors reply:
*We will add the posterior spread to show the goodness of fit. To explore and compare with other low-dimensional models than the Langevin model is beyond the scope of this paper*

Here I am not sure what has been added. The densities shown in Fig. 7 were already present in the previous version, and do not really help here, partly because the distributions have been inflated by a factor nu/2. What is interesting are differences of the best fit relative to the data. Maybe the authors could at least quantify the "goodness of fit" of the Langevin model to the different data sets and compare them, in order to point out where the model has issues capturing the EMIC behavior (apart from the resurgence of the AMOC).

**2.**
Regarding the best fit parameters, the authors have responded:
*The most likely value is the mean of the posterior distribution. This does assume that the posterior distributions are uni-modal. We will discuss this in the text.*

The most likely value is not the mean, but the mode of the distribution. This makes an especially large difference for skewed distributions and bi-modal distributions. Can the authors show that the posterior distributions are uni-modal? This needs to be better clarified in the manuscript. Furthermore, I assume they use the marginal posterior distributions of the individual parameters.
The corresponding segment in the revised manuscript reads:
*"The parameter values of these distributions are the means of the posterior distributions."*
This should be rewritten at least: "As best fit parameters, we choose the mean values of the marginal posterior distributions."

**3.**
Regarding the normalization factor C, the authors responded:
*The reviewer is correct, we added. " but can be computed numerically (and therefore used as a likelihood function in the next section)"*

Now it reads as if C can be used as likelihood function, rather than P. Please correct.

**4.**
P16L272: "These six parameters are sufficient to describe the abrupt collapse of the AMOC that leads to a hysteresis loop under varying freshwater forcing."

This statement should be made more precise. It is not the collapse that leads to a hysteresis loop, both are a result of the same underlying mechanisms. Maybe the authors can say that

the parameters are sufficient to describe an abrupt collapse of the AMOC, as part of a hysteresis loop under varying freshwater forcing.

**5.**
P16L272: "… and we did not attempt to obtain an accurate fit of that part of the hysteresis loop."

I think it would be fair to say that with the model at hand it is simply not possible to fit the data, since while asymmetric branches of the hysteresis loop may be obtained, the noise model used is symmetric. Also, it is not clear what is meant by "the resurgence of the AMOC is not the same as the collapse". In a way, when invoking just one positive feedback, it is.
This explanation could be merged with the two very short paragraphs that follow.

Technical corrections:

Abstract: "Steady-state collapse trajectories of the AMOC…."
Maybe better terminology can be found? I don't think that "steady-state trajectories" are a very intuitive concept, and it is not clear what is meant here. Maybe simply say "hysteresis diagrams" or the like.
I also suggest to rewrite: "Differences in between the climate models studied here are mainly due to..."

P4L74: Remove "studied mainly qualitatively in connection with the Langevin equation".

P4L77: More precisely, the third order term and the fourth order coefficient can be eliminated.

P11L198: "Conceptually, this is what an MCMC…"

P13L249: " Our main goal is to model the transition from the on-branch to the off-branch…"

P13L249: "Also, because we assume that other dynamics govern the lower branch and, our simple model has would need to be extended to account for those dynamics

P18L338: … but at present in it is unknown ….

---

## Author Response (AR2)

[revised manuscript text omitted]

Comments to the Author:
Dear authors,
the manuscript has improved and requires only minor revisions.

1) please follow the suggestions of the referee, and provide a point-by-point answer to the comments.

We thank the editor for the additional comments and suggestions for improvement. Below our responses.

2) My own evaluation and following comments can improve the quality of the paper.

a) page 2, lines 25-28: Referencing to double well potential and dynamics of abrupt climate shifts. I think the method and application of Livina and others were first described here: Kwasniok, F., and G. Lohmann, 2009: Phys. Rev. E, doi: 10.1103/PhysRevE.80.066104
This paper predates the reference to Livina et al (2010) and relates to ice core data. A stochastically driven motion in a 4$^{th}$ order polynomial potential is used to model glacial state changes on the millennial scale. We have included it as an additional reference.

b) I think that the statement
"1 This algorithm has been implemented in many software packages."
could be avoided as footnote. You could include this statement in the full text and provide references.
We have included references to two implementations in the main body of the text.

c) section4, lines 270ff, 282 ff. The Langevin model is a simplified view and many features like the gyres are missing. In Prange et al. (2003) it has been shown that even 2D and 3D models can have a substantial different dynamics because of the missing dimension. I guess that this is even more true for the 0-d model. In 3 D models, a reversed flow as off mode does not exist (see e.g. on Fig. 9 of their paper) due to the gyres (section 4b). THis might be also related to your statements lines 250 ff on page 13, and lines 260 ff on page 14, lines 274 and 283 on page 275. The salinity advection mechanism is indeed not the mechanism for the off state. If feel that the speculation with the metric (lines 284-286) could take this into account. The way it is written now is somehow misleading.
Indeed, the statements on lines 250 and 260 relate to the differences in upper and lower branches. A lower dimensional model necessarily needs to ignore geometry that can be found in higher dimensional ones. We have added a remark to highlight this. "It is unclear to what extent the models discussed here develop a reversed overturning circulation which can arise in 3D models (Weijer and Dijkstra, 2001; Yin and Stouffer, 2007), but which can also be suppressed by atmospheric feedbacks (Yin and Stouffer (2007); however, see also Mecking et al. (2016)), and strongly affected by gyre dynamics (Prange et al., 2003). These effect are not captured by the simple Langevin model proposed here, but at present it is still unclear to what extent these effects are essential in capturing the first order stability properties of the AMOC. In each case, there is no obvious..."

d) page 17, line 322: This statement is not correct: "it is not clear how long the models in Rahmstorf et al. (2005) were integrated per freshwater forcing value". This value is given in the paper. "The rate of change of the freshwater input was 0.05 Sv per 1,000 model years." This information could be easily included.
We added the integration times and forcing value increment value.

e) page 18, lines 327 ff: The statement "As noted by Gent (2018), the hysteresis behaviour has not been investigated fully in models of greater complexity than EMICs" is probably not true. See some

examples by Mikolajewicz (with a higher freshwater rate in the hysteresis) and other colleagues like Zhang in 2017. Just reformulate.

We have reformulated and added the following references:
Schiller, A., Mikolajewicz, U., & Voss, R. (1997). The stability of the North Atlantic thermohaline circulation in a coupled ocean-atmosphere general circulation model. *Climate Dynamics*, **13**(5), 325– 347.
Zhang, X., Knorr, G., Lohmann, G., & Barker, S. (2017). Abrupt North Atlantic circulation changes in response to gradual $CO_2$ forcing in a glacial climate state. *Nature Geoscience*, **10**(7), 518.
It should be noted that the model used in these studies have a simplified physics (such as the large scale geostrophic model) and do not have complexity found in current CMIP class models.

f) page 18, line 337 ff: In coupled models M_ov is just describing the oceanic feedback, it implicitly assumes that the circulation within the Atlantic is a second order effect. In truely coupled atmosphere-ocean model is seems that the atmosphere has a similar order of magnitude affecting the freshwater (e.g. Lohmann 2003; Ackermann et al. (2020, GRL) and others. In this direction also the Liu et al. work is interesting that in not flux-corrected coupled GCMs the salinity bias is so strong providing a more monostable behaviour.:w

The atmosphere can be of great importance for AMOC recovery as Lohmann (2003) and Ackermann et al. (2020) show. Work by Liu et al. Shows the climate models to bee too stable due to salinity biases. We have included these references and expanded the discussion on this point.

Additional remark: We have received the original data files from Rahmstorf et al. (2005) and performed the same calculations as presented in the draft versions of the paper on this data set. Qualitatively the results are the same, and our conclusions are unchanged. We have also regridded these values on a uniform set of forcing values to aid comparison; this also sped up the calculations. There have been some changes in the graphics and fitted parameter values.

Thank you for the opportunity to review this manuscript again. While it would have been nice for the authors to address several points more explicitly, which would have helped to increase the scope of the paper, the corrections helped to make the text more readable and make the purpose of the paper more clear.

We thank the reviewer for the additional comments and suggestions. Below are our responses. Additionally, we have received the original data files from Rahmstorf et al. (2005) and processed these after regridding them onto the same freshwater forcing range. The graphics and fit values have changed, but the qualitative and, therefore, our conclusions have not.

I have a few additional comments:
1.
Regarding how they can substantiate that the model successfully captures the dynamics, the authors reply:
We will add the posterior spread to show the goodness of fit. To explore and compare with other low-dimensional models than the Langevin model is beyond the scope of this paper

Here I am not sure what has been added. The densities shown in Fig. 7 were already present in the previous version, and do not really help here, partly because the distributions have been inflated by a factor nu/2. What is interesting are differences of the best fit relative to the data. Maybe the authors could at least quantify the "goodness of fit" of the Langevin model to the different data sets and compare them, in order to point out where the model has issues capturing the EMIC behavior (apart from the resurgence of the AMOC).
We have removed the spreads in the table and added a column with the root-mean-square deviation calculated on the upper branch up to the collapse points to give an indication of the goodness of fit.

2.
Regarding the best fit parameters, the authors have responded:
The most likely value is the mean of the posterior distribution. This does assume that the posterior distributions are uni-modal. We will discuss this in the text.

The most likely value is not the mean, but the mode of the distribution. This makes an especially large difference for skewed distributions and bi-modal distributions. Can the authors show that the posterior distributions are uni-modal? This needs to be better clarified in the manuscript. Furthermore, I assume they use the marginal posterior distributions of the individual parameters.
The corresponding segment in the revised manuscript reads:
"The parameter values of these distributions are the means of the posterior distributions."
This should be rewritten at least: "As best fit parameters, we choose the mean values of the marginal posterior distributions."
The reviewer is correct: neglecting "marginal" is sloppy terminology. Corrected as suggested. Although MCMC posterior distributions are typically unimodal, we do not check for this property though.

3.
Regarding the normalization factor C, the authors responded:
The reviewer is correct, we added. " but can be computed numerically (and therefore used as a likelihood function in the next section)"
Now it reads as if C can be used as likelihood function, rather than P. Please correct.
We changed "...therefore used as a likelihood function…" to "…can therefore used as a factor in the likelihood function…"

4.
P16L272: "These six parameters are sufficient to describe the abrupt collapse of the AMOC that leads to a hysteresis loop under varying freshwater forcing."
This statement should be made more precise. It is not the collapse that leads to a hysteresis loop, both are a result of the same underlying mechanisms. Maybe the authors can say thatthe parameters are sufficient to describe an abrupt collapse of the AMOC, as part of a hysteresis loop under varying freshwater forcing.
Rewritten "...the AMOC that leads to a hysteresis loop …" as "...the AMOC as part of a hysteresis loop..."

5.
P16L272: "... and we did not attempt to obtain an accurate fit of that part of the hysteresis loop."
I think it would be fair to say that with the model at hand it is simply not possible to fit the data, since while asymmetric branches of the hysteresis loop may be obtained, the noise model used is symmetric. Also, it is not clear what is meant by "the resurgence of the AMOC is not the same as the collapse". In a way, when invoking just one positive feedback, it is.
This explanation could be merged with the two very short paragraphs that follow.
Moved sentence down two paragraphs and rewritten. Removed sentence at P16L279.

Technical corrections:
All suggestions and corrections below accepted.

Abstract: "Steady-state collapse trajectories of the AMOC...."
Maybe better terminology can be found? I don't think that "steady-state trajectories" are a very intuitive concept, and it is not clear what is meant here. Maybe simply say "hysteresis diagrams" or the like.
Agreed: this terminology is not clear. Replaced with "Hysteresis diagrams" as suggested.
I also suggest to rewrite: "Differences in between the climate models studied here are mainly due to…"
Replaced with " Differences between the climate models studied here are mainly due to..."

P4L74: Remove "studied mainly qualitatively in connection with the Langevin equation".
removed

P4L77: More precisely, the third order term and the fourth order coefficient can be eliminated.
rewritten

P11L198: "Conceptually, this is what an MCMC…"
corrected

P13L249: " Our main goal is to model the transition from the on-branch to the off-branch…"
corrected

P13L249: "Also, because we assume that other dynamics govern the lower branch and, our simple model has would need to be extended to account for those dynamics
corrected

P18L338: ... but at present in it is unknown .…
corrected